# Single-handed supramolecular double helix of homochiral bis(*N*-amidothiourea) supported by double crossed C−I⋯S halogen bonds

Xiaosheng Yan [1], Kunshan Zou[1], Jinlian Cao[1], Xiaorui Li[1], Zhixing Zhao[1], Zhao Li[1], Anan Wu[2], Wanzhen Liang[2], Yirong Mo[2,3] & Yunbao Jiang[1]

The natural DNA double helix consists of two strands of nucleotides that are held together by multiple hydrogen bonds. Here we propose to build an artificial double helix from fragments of two strands connected by covalent linkages therein, but with halogen bonding as the driving force for self-assembling the fragments to the double helix. We succeed in building such a double helix in both solution and solid state, by using a bilateral *N*-(*p*-iodobenzoyl) alanine based amidothiourea which in its folded *cis*-form allows double and crossed C−I⋯S halogen bonds that lead to right- or left-handed double helix when the two alanine residues are of the same L,L- or D,D-configuration. The double helix forms in dilute $CH_3CN$ solution of the micromolar concentration level, e.g., 5.6 μM from 2D NOESY experiments and exhibits a high thermal stability in solution up to 75 °C, suggesting cooperative and thereby strong intermolecular double crossed halogen bonding that makes the double helix stable. This is supported by the observed homochiral self-sorting in solution.

[1] Department of Chemistry, College of Chemistry and Chemical Engineering, The MOE Key Laboratory of Spectrochemical Analysis and Instrumentation, and iChEM, Xiamen University, 361005 Xiamen, China. [2] Department of Chemistry, College of Chemistry and Chemical Engineering, Xiamen University, 361005 Xiamen, China. [3] Department of Chemistry, Western Michigan University, Kalamazoo, MI 49008, USA. Correspondence and requests for materials should be addressed to Y.J. (email: ybjiang@xmu.edu.cn)

The DNA double helix, well-known since its discovery by Watson and Crick in 1953[1], consists of two complementary polynucleotide strands that are connected by multiple hydrogen bonds, with additional stabilization by π–π stacking, between the nucleobase pairs that are covalently attached to the strands. Due to the significant functionality of DNA double helix, including information storage and transfer, building artificial double helices has been a challenging subject of great interest, in particular in solution. Two approaches have been developed. The first is to intertwine two oligomeric or polymeric strands into double helices by inter-strand non-covalent interactions, such as hydrogen bonding[2–4], metal coordination (Cu⁺ to pyridine units)[5–7], amidinium-carboxylate salt bridges[8–10], and aromatic-aromatic interactions between aromatic oligoamides[11,12] or oligoresorcinols strands[13,14]. This is in principle similar to the mechanism leading to DNA. The second is a bottom-up approach from small molecules through multiple noncovalent interactions, i.e., hydrogen bonding, hydrophobic interaction and π–π stacking[15–21]. For example, Yagai et al.[20] reported hierarchical assembly of a chiral azobenzene dimer bearing a 3,4,5-(tridecyloxy)xylylene linker into supramolecular double-stranded helices. In nonpolar solvent MCH at 0 °C, the azobenzene molecules at 0.3 mM first form helical stacked nanotubes via noncovalent π–π stacking and hydrogen bonding, which further twist into single helical strand of staircase supramolecular structures, followed by the intertwining of two adjacent strands into double helices after aging at 0 °C for days. However, a double helix formed directly from synthetic small molecules via intermolecular interactions remains to be explored.

In the DNA double helix, the two strands are intertwined by multiple hydrogen bonds between the complementary nucleobases, respectively, from the two strands in which the nucleobases are linked by the covalent phosphate backbone (Fig. 1a, left)[1]. We therefore envisage that by replacing the multiple hydrogen bonds in DNA double helix with covalent linkages, but using multiple intermolecular interactions instead of covalent bonds to link the small molecules into supramolecular strands, we will be able to build an artificial double helix (Fig. 1a, right). In our case, crossed double non-covalent interactions between two neighboring molecules are the driving force. Bilateral molecules bearing complementary bidentate binding sites can be potential building blocks.

We recently reported an alanine based bis($N$-amidothiourea), L,L-**AI** or D,D-**AI** (Fig. 1b, right), which can form a single-stranded supramolecular helix[22]. The **AI** molecules (**AI**s) take the extended *trans*-form in the helix (Fig. 1b, left), therefore allowing a head-to-tail C−I···π halogen bonding that supports the single-stranded supramolecular helix, in which the helicity of the two helical β-turn structures is well propagated. Inspecting the structure of the disfavored folded *cis*-**AI**, we envision that it approaches the desired structure for our design of an optimal framework with crossed double interactions between neighboring two molecules to lead to double helix. In the *cis*-form of **AI**, the geometry and the distance of the halogen bonding donor, the I-atom, with respect to the acceptor atom, S-atom or O-atom, are both not optimal for potential C−I···S or C−I···O halogen bonding[23–29].

In this work, we anticipate that moving the S-atom closer towards the central benzene ring while allowing a similar helical β-turn structure to remain will make it possible to arrive at the optimal structural motif. Since the helicity of the helical β-turn fragments in **AI** plays an important role in supporting the supramolecular single-stranded helix[22], we decide to design its "tail-to-tail" derivatives, L,L-**IA**, D,D-**IA**, and L,D-**IA** (Fig. 1c, right), in that each **AI** molecule adopt the "head-to-head" structure. Despite slightly less favorable from the DFT calculations, the *cis*-form **IA** (Fig. 1c, left) affords a conformation that the two S-atoms are in the optimal range, and looks like the constitutional component shown in the right of Fig. 1a. Ultimately, the *cis*-form **IA** molecules are likely to form crossed double halogen bonds with two I-atoms from an adjacent **IA** molecule, leading to a supramolecular double helix (Fig. 1c, left). We find that the crossed double C−I···S halogen bonds between *cis*-form homochiral bilateral $N$-amidothioureas (e.g., L,L-**IA** or D,D-**IA**), as the intra-strand noncovalent interactions, occur to support the single-handed supramolecular double helix in both the solid state and more significantly in extremely dilute CH₃CN solution, while the central *p*-phenylenediamine moiety acts as the inter-strand covalent linkage. To our knowledge, this establishes a unique approach to the supramolecular double helix formed in both solid state and dilute solution from synthetic small molecules via intermolecular noncovalent interactions, i.e., the halogen-bonding.

## Results

**Crystal structures.** I-substituted bilateral $N$-amidothioureas **IA**s (L,L-/L,D-/D,D-**IA**, Fig. 1c right) were synthesized via procedures described in Supplementary Figs. 1 and 2. Single crystals were obtained by slow vapor diffusion of diethylether into DMF solutions of **IA**s (for crystallographic data, see Supplementary Table 1). DFT-optimized structures of L,L-**IA** in gas phase show that the *trans*-form is favored over *cis*-form by 8.37 kJ mol⁻¹ (Fig. 1c left). However, X-ray crystal structure analysis revealed that L,L-**IA** only adopts one *cis*-form containing two identical helical β-turn structures, **β1** and **β2** (Fig. 2a)[30]. The **β1** and **β2** turns are both of type II, with same structural parameters (Supplementary Table 2)[31,32].

To understand the existence of the otherwise thermodynamically disfavored *cis*-form L,L-**IA** in the crystals, molecular packing effect was examined. Two I-atoms ($I_1$ and $I_2$) in one L,L-**IA** molecule are crossed in geometry, interacting along the *c*-axis, respectively, with $S_2$ and $S_1$ atoms from the adjacent L,L-**IA** molecule, by two C−I···S halogen bonds, **XB1** and **XB2** (Fig. 2b). The double C−I···S halogen bonds are the same in structural parameters (Supplementary Table 3), being 3.764(3) Å in length that is shorter than the sum of the van der Waals radii of I and S atoms (3.780 Å)[33], whereas the angle of the halogen bond is 156.3(3)°, being almost linear. The bond length is longer than the usual I···S contacts, such as those in crystalline iodinated dithiole-2-thiones and thiazole-2-thiones that range from 3.2 to 3.4 Å[27], but it is comparable to those in the classic crystals of I-substituted tetrathiafulvalene derivatives with $I_3^-$ or $I_2$ that range between 3.7–3.9 Å[28]. The longer length observed here may result from the balance of the bond lengths and angles, since shorter lengths will lead to smaller angles in the crossed geometry with two I···S contacts. In the crystal structure, each *cis*-form L,L-**IA** molecule is involved in four C−I···S halogen bonds, yet the calculated interaction energy of one C−I···S halogen bonding in the dimer of *cis*-L,L-**IA**, 23.89 kJ mol⁻¹ (Supplementary Table 3)[28], is already high enough to compensate the energy penalty (8.37 kJ mol⁻¹) of the less stable *cis*-form L,L-**IA** over its *trans*-form. This calculated interaction energy is comparable to that (25.02 kJ mol⁻¹) of the I···S interaction in the classic crystal of 1,3-dithiole-2-thione-4-carboxylic acid with $I_2$[34]. This considerably high interaction energy can at least be partly attributed to the double crossed geometry of the halogen bonding in the double helical structure where there is considerable secondary electrostatic interaction[35,36], since in the simulated dimer of unilateral analogs of bilateral L,L-**IA** (half structure of the dimer of *cis*-L,L-**IA**) with only one C−I···S halogen bond, the interaction energy decreases dramatically to 15.04 kJ mol⁻¹ (Supplementary Fig. 3). The

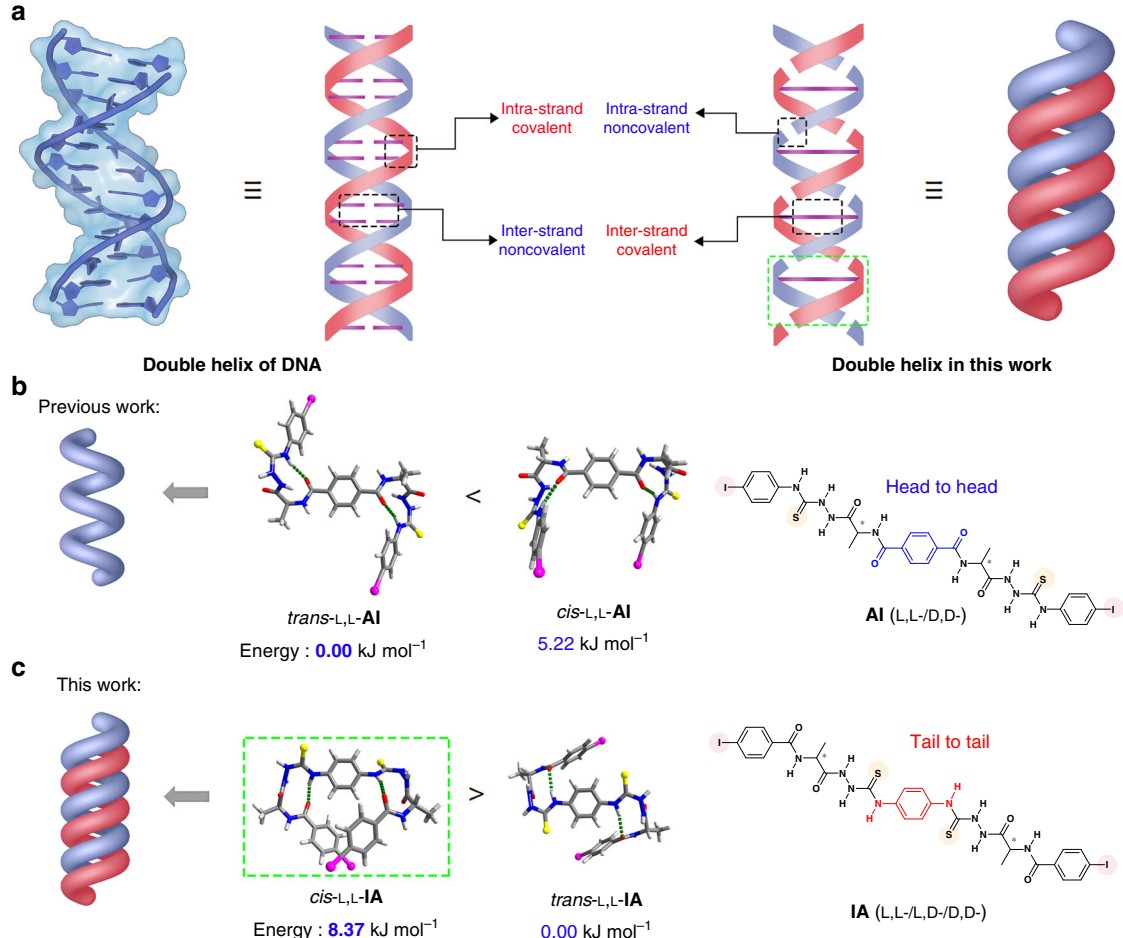

**Fig. 1** Schematic illustration of double helix and molecular design. **a** Schematic diagram of the double helix of DNA and that built in this work, featuring intra-strand noncovalent and inter-strand covalent interactions in the latter. **b** DFT-optimized structures and corresponding energies of *trans*-L,L-**AI** and *cis*-L,L-**AI** in gas phase and molecular structures of **AI**. The single-stranded supramolecular helix is formed from *trans*-L,L-**AI**[22]. **c** DFT-optimized structures and corresponding energies of *trans*-L,L-**IA** and *cis*-L,L-**IA** in gas phase and molecular structures of **IA** in this work. The supramolecular double helix is formed from *cis*-L,L-**IA**. The asterisks in the structures indicate the chiral carbons. Dashed green lines highlight intramolecular hydrogen bonds (IHBs) that represent β-turn structures. Method: DFT B3LYP with the 6–311 G** basis set for C, H, O, N, and S atoms, and LANL2DZ for I atoms

infinite halogen bonds therefore hold the *cis*-form L,L-**IA** molecules into 1D-superstructure along the *c*-axis (Fig. 2c). It is significant to observe that in the 1D-superstructure of L,L-**IA**s, the crossed double C−I···S halogen bonds link the helical β-turn fragments[37] into two *P*-helical strands (Fig. 2d). The two strands are intertwined by the central covalent *p*-phenylenediamine linkage in L,L-**IA**s to afford the *P*-double helix (Fig. 2e). The single-handed supramolecular double helix is 2.60 nm in pitch while 1.18 nm in diameter (Fig. 2c, e). Compared to the natural double helix of DNA that is supported by intra-strand covalent phosphate backbones and inter-strand noncovalent hydrogen bonds, the *P*-double helix of *cis*-L,L-**IA**s results from intra-strand noncovalent halogen bonds and inter-strand covalent *p*-phenylenediamine linkage (Supplementary Table 4)[21]. It therefore represents a unique protocol to build double helix. In addition to the four C−I···S halogen bonds (Fig. 2c), one *cis*-form L,L-**IA** is also involved in eight C=O···H−N hydrogen bonds with four surrounding molecules (Supplementary Fig. 4), leading to 3D-supramolecular architectures (Supplementary Fig. 5).

The occurrence of the halogen bond in the solid state of L,L-**IA** was further supported by Raman and infrared spectroscopic data. Compared with the synthetic material that has not the thiourea moiety, C−I bond in L,L-**IA** showed a lower Raman shift at *ca.* 171 cm$^{-1}$ (Supplementary Fig. 6). This is consistent with the

involvement in the halogen bonding of the I$^-$atoms in L,L-**IA**[38]. The stretching vibration of C=S double bond in the thiourea moiety appeared in general around 1530 cm$^{-1}$ (Supplementary Fig. 7)[39]. We observed a red-shifted band in L,L-**IA**, compared to the control compounds L,L-**FA**, L,L-**ClA** and L,L-**BrA** (Fig. 3a) that contain halogen atoms of −F, −Cl, and −Br of lower efficiency of halogen bonding, thus supporting the C−I···S halogen bonding between L,L-**IA** molecules in the solid state, as also revealed by the crystal structures[40].

The enantiomer of L,L-**IA**, D,D-**IA** (Fig. 1c right), also adopts the *cis*-form in its crystal yet with two identical type-II′ β-turns at both sides of the compound (Supplementary Table 2), forming the *M*-double helix that is supported by crossed double C−I···S halogen bonds as well (Supplementary Fig. 8). The *P*-double helix from L,L-**IA** and the *M*-double helix from D,D-**IA** are structurally mirror images (Fig. 2e), indicating that the helicity of the supramolecular double helix is determined by the absolute configuration of the alanine residues. The heterochiral derivative, L,D-**IA** (Fig. 1c right), however, exists in the *trans*-form. X-ray crystal structure shows that the two sides of the L,D-**IA** molecule are symmetric (*P*-1 space group), containing a type-II (with L-alanine residue) and a type-II′ (D-alanine residue) β-turns (Supplementary Table 2). Instead of the C−I···S halogen bonds observed in the crystals of homochiral L,L-**IA** and D,D-**IA**, two

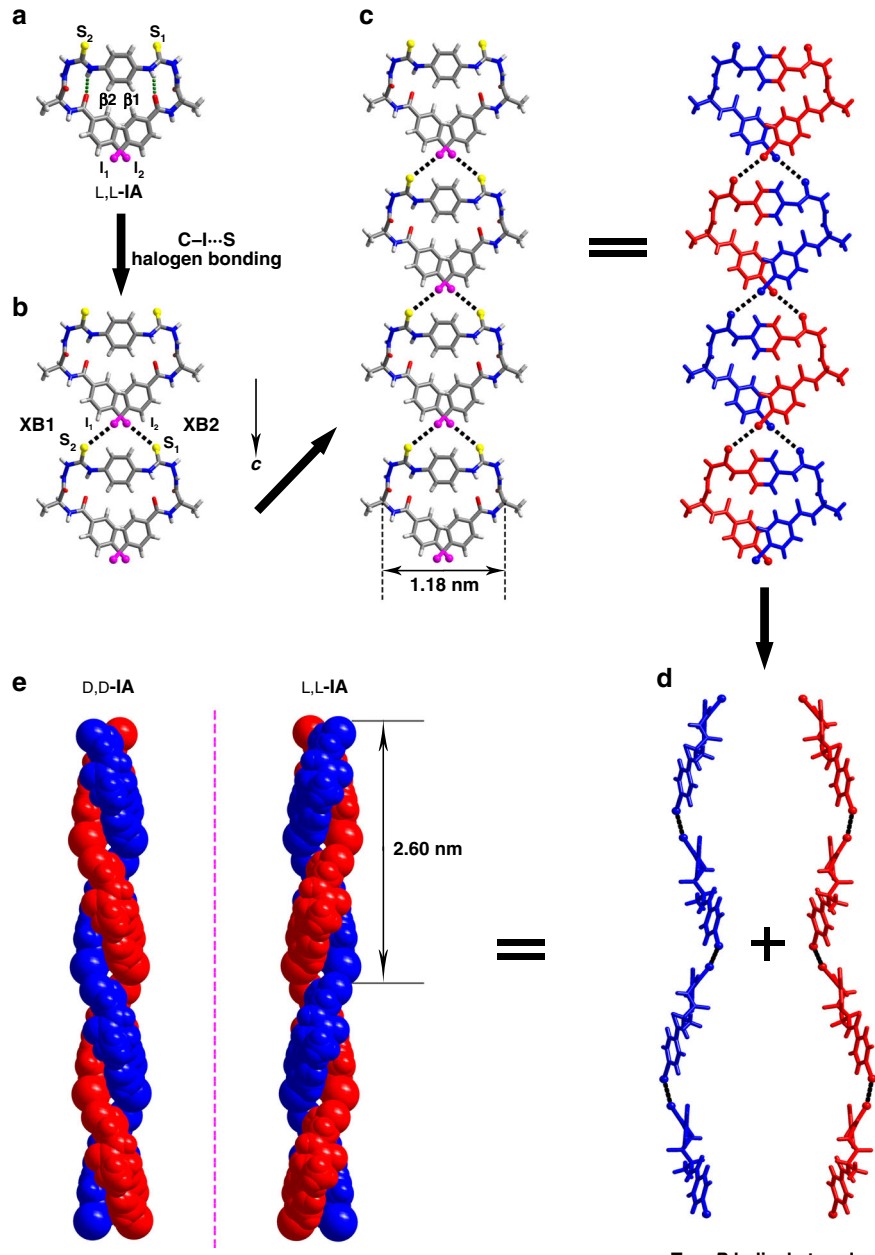

**Fig. 2** Single-handed supramolecular double helix from L,L-**IA** and D,D-**IA**. **a** X-ray crystal structure of L,L-**IA** showing *cis*-form with β-turn at each side. Dashed green lines highlight the IHBs that form the β-turn structures **β1** and **β2**. **b** Crossed double C−I···S halogen bonds between two adjacent *cis*-form L, L-**IA** molecules. Dashed black lines highlight two C−I···S halogen bonds **XB1** and **XB2**. **c** 1D superstructure of *cis*-form L,L-**IA** molecules through C−I···S halogen bonding along *c*-axis. **d** Two *P*-helical strands in 1D superstructure of L,L-**IA**. **e** *M*-double and *P*-double helices formed, respectively, from the enantiomeric D,D-**IA** and L,L-**IA**. For clarity reasons, one side in the bilateral D,D-**IA** and L,L-**IA** molecules is depicted in blue, while the other side is depicted in red

parallel C−I···O halogen bonds (Supplementary Table 5) were found between the L-side of one L,D-**IA** molecule and the D-side of the adjacent L,D-**IA** molecule, resulting in a zigzag 1D-superstructure (Supplementary Fig. 9). These results demonstrate that molecular chirality plays a decisive role in the super-structures, leading to single-handed double helix from homo-chiral L,L-**IA** or D,D-**IA** molecules, whereas non-helical zigzag structure from the heterochiral L,D-**IA** derivative.

**Supramolecular double helix formation in CH₃CN solution.** We next examined whether the C−I···S halogen bonds supported supramolecular double helices survive in the solution phase. For

comparison, control compounds L,L-**XA** (X = **H**, **F**, **Cl**, **Br**, Fig. 3a) were examined too in order to illustrate the role of the I-substituents in L,L-**IA**. DFT calculations showed that all of them favor the *trans*-form over the *cis*-form, by energies close to that of L,L-**IA** (Supplementary Table 6). Different from those of the control compounds, L,L-**IA** in CH₃CN exhibited red-shifted absorption and, in particular, dramatically enhanced CD signals with Cotton effects at 342, 291 and 259 nm, respectively (Fig. 3b). The g-factor of L,L-**IA** is up to $-1.6 \times 10^{-2}$ at 291 nm (Supplementary Fig. 10), implying the formation of helical super-structures in the highly dilute CH₃CN solution of 5 μM[41]. The positive Cotton effect of L,L-**IA** at long wavelength 342 nm

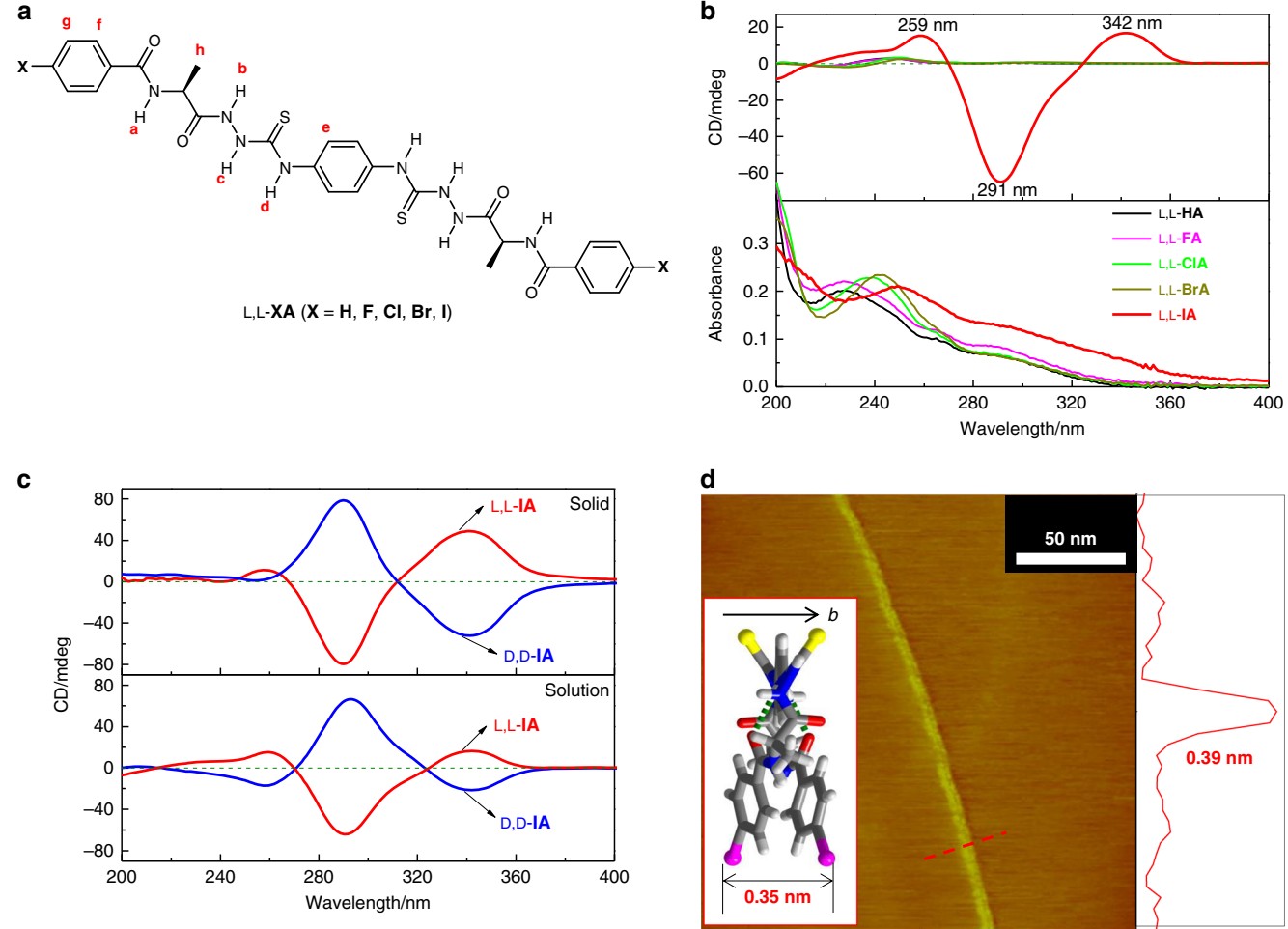

**Fig. 3** Absorption and CD spectra. **a** Molecular structures of bilateral *N*-amidothioureas L,L-**XA** (X = **H**, **F**, **Cl**, **Br**, **I**) with labeled protons. **b** Absorption and CD spectra of L,L-**XA** in CH$_3$CN. [L,L-**XA**] = 5 μM. **c** CD spectra of L,L-**IA** and D,D-**IA** in CH$_3$CN solution and the solid state. For solution, [**IA**] = 5 μM, while the concentration of the solid CD sample is about 1.0 mg/400 mg KCl. **d** STM height image of L,L-**IA** at the CH$_3$CN-HOPG interface with height profile along the red dashed line. Inset is the thickness of single *cis*-L,L-**IA** molecule along *b*-axis revealed by the crystal structures. [L,L-**IA**] = 1 μM

indicated the *P*-helical structures in CH$_3$CN[42], in agreement with the *P*-double helices identified in the crystal structures of L,L-**IA** (Fig. 2e). The existence of supramolecular polymeric species of L, L-**IA** in CH$_3$CN solution was confirmed by dynamic light scattering (DLS, Supplementary Fig. 11), which showed large species of diameters of *ca*. 255 nm[43–45]. In contrast, DLS data suggested that those control compounds L,L-**XA**s (X ≠ I) existed in monomer forms in CH$_3$CN (Supplementary Fig. 11), as also indicated by their absorption and CD spectra (Fig. 3b) and, presumably due to the lower efficiency of halogen bonding of –F, –Cl, and –Br compared to that of –I[29]. Moreover, the hydrodynamic diameters of L,L-**IA** in CH$_3$CN measured at different scattering angles varied significantly (Supplementary Fig. 12), indicating a pronounced anisotropy of the existing species[43]. This is consistent with the long polymeric chain structures that are revealed by the crystals.

CD spectra of L,L-**IA** and D,D-**IA** in CH$_3$CN solutions were found to be similar to those in the solid state (Fig. 3c), likely indicating a similar helical structure of L,L-**IA** or D,D-**IA** molecules in CH$_3$CN solution to that in the solid state. The CD spectral profile of L,L-**IA**, with the first positive, the second negative and the third positive Cotton effects that are indicative of the *P*-helicity (Fig. 3b)[42], is in agreement with the *P*-double helix identified for L,L-**IA** in the crystals, and therefore concluded in CH$_3$CN solution as well. For other L,L-**XA**s (X ≠ I) that exist in

the monomer forms in CH$_3$CN while to some extents in stacked forms in the solid state, their CD spectra differ in the solutions from those in the solid (Supplementary Fig. 13). To support the double helical structure of L,L-**IA** in solution phase, we calculated the solution phase CD spectra of the double helical structures and the monomers as well. While the calculated CD spectra of the monomeric *trans*-L,L-**IA** and *cis*-L,L-**IA** in CH$_3$CN solution are very different from the experimentally observed CD spectrum of L,L-**IA** in CH$_3$CN (Supplementary Fig. 14), those calculated for the crossed double halogen-bonded dimer and trimer of *cis*-L,L-**IA** are similar to the experimental CD spectrum, featuring the first positive, the second negative and the third positive Cotton effects that are indicative of the *P*-helicity. Therefore, the similarity of the CD spectrum of L,L-**IA** in CH$_3$CN solution to that in the solid state of the double helical structure would be more strongly to indicate the double helical structure in solution phase. SEM images of air−dried samples of L,L-**IA** and D,D-**IA** in CH$_3$CN showed ordered rod-shaped aggregates, whereas amorphous blocks for L,L-**HA**, L,L-**FA**, L,L-**ClA**, and L,L-**BrA** (Supplementary Fig. 15), again demonstrating the significance of the I-substituents in defining the superstructures of L,L-**IA** and D,D-**IA**. In particular, the STM images of L,L-**IA** and D,D-**IA** at the CH$_3$CN-highly oriented pyrolytic graphite (HOPG) interface showed long polymeric chains (Fig. 3d and Supplementary Fig. 16), a shape of anisotropy that is consistent with the varying

diameters of L,L-**IA** measured at different scattering angles (Supplementary Fig. 12)[43]. The heights of the chains were determined to be 0.39 nm and 0.37 nm, respectively, for L,L-**IA** and D,D-**IA** from the height profiles, agreeing with the thickness of one *cis*-form L,L-**IA** and D,D-**IA** along *b*-axis (0.35 nm) deduced from the crystal structures. These results well support the 1D supramolecular polymeric chain structures in the solution phase, as those shown in the crystal structures (Fig. 2). Hence L,L-**IA** and D,D-**IA** in dilute $CH_3CN$ solution forms the single-handed supramolecular double helices as well.

CD signal of L,L-**IA** in $CH_3CN$ at 25 °C was found to vary with its concentration sigmoidally, giving an extremely low critical aggregation concentration of 0.08 μM (Supplementary Fig. 17). A high equilibrium constant $K_e$ of $1.25 \times 10^7$ $M^{-1}$ at 25 °C was calculated, according to the assumption that the critical aggregation concentration is close to $K_e^{-1}$[46]. The estimated $K_e$, higher than those of the helical columnar stacks of classical $C_3$-symmetrical benzene-1,3,5-tricarboxamides of $10^5$ $M^{-1}$ level, is ascribed to the high cooperativity of the crossed double halogen bonds[47]. Meanwhile, the CD signal almost did not drop when the $CH_3CN$ solution was heated from 20 to 75 °C, implying a high thermal stability of the supramolecular structures of L,L-**IA** in solution (Supplementary Figs. 18–20), higher than that of the single-stranded supramolecular helix formed from L,L-**AI** in $CH_3CN$ solution whose CD signal started to drop at 35 °C (Supplementary Fig. 21)[22]. Concentration dependent DLS data of the solutions showed little change in the size of the polymeric species upon varying concentration from 0.5 to 5 μM (Supplementary Fig. 22). This observation suggested that the supramolecular polymeric forms of L,L-**IA** in $CH_3CN$ exist already at very low concentration and the increase in the solution concentration over 0.5 to 5 μM mainly results in the increase in the number but not the length of the supramolecular species. After standing for 7 days at room temperature, the solution of L,L-**IA** in $CH_3CN$ shows little change in the DLS data (Supplementary Fig. 23). Decrease in the size of the polymeric species was observed when the solution was heated from 25 to 75 °C (Supplementary Fig. 24), yet the fact that large supramolecular species (*ca.* 100 nm) exists at 75 °C again indicated that the superstructure remains stable at high temperature. These results demonstrated that the intermolecular interactions are cooperative thereby tightly hold L,L-**IA** molecules into the supramolecular helical structures in $CH_3CN$, even at extremely low concentration, standing for several days and at high temperature.

This high strength of the double helix was also reflected in the facts that the absorption and CD spectra of L,L-**IA** in $CH_3CN$ were not affected by 5.0 eq halogen anions $Cl^-$, $Br^-$, and $I^-$ (Supplementary Figs. 25–28) that could act as acceptor of halogen bonding[48–51]. This means that the supramolecular double helix in $CH_3CN$, formed via intermolecular double crossed C−I···S halogen bonding, is so strong that it remains even in the presence of 5.0 eq highly competitive $I^-$. Introduction of the halogen anions at high concentration of up to 1000 eq, however, results in a gradual drop of the values of the CD signals of the supramolecular helical structures (Supplementary Fig. 29). This means the damaging of the double helical structure by halogen anions at much higher concentration, with the capability of damaging in the order of $I^- > Br^- > Cl^-$ (Supplementary Fig. 30). This order is opposite to that of the hydrogen bonding ability of these anions with the thiourea moiety in L,L-**IA**, but it is consistent with the halogen-bonding ability of the halogen anions[49]. This could serve to probe the role of the halogen bonding in supporting the supramolecular helical structures of L, L-**IA** in $CH_3CN$ solution.

The halogen bonding nature of the intermolecular interactions was also confirmed by the solvent effect upon introducing into $CH_3CN$ the solvents of increasing basicity, $CHCl_3$, THF, and $H_2O$. With 20% volume fraction of low basicity $CHCl_3$ in $CH_3CN$, CD spectrum of L,L-**IA** remained unchanged, whereas it reduced slowly in the presence of 20% electron-donating solvent THF, and the reducing became fast when $H_2O$ was added (Supplementary Figs. 31 and 32). The final CD spectrum of L,L-**IA** in $CH_3CN$ containing 20% $H_2O$ standing for long time was similar to those of the control L,L-**XA**s (X = H, F, Cl, Br, Supplementary Fig. 33), suggesting the transformation of the supramolecular double helix of L,L-**IA** in $CH_3CN$ to the monomer form of L,L-**IA** in $CH_3CN/H_2O$. This phenomenon proves the halogen bonding nature of the intermolecular interactions[52]. The CD signals of L,L-**IA** in $CH_3CN/H_2O$ drop more slowly than those of L,L-**AI** (Supplementary Fig. 34)[22], also indicating a higher strength of the double helix of L,L-**IA** than that of the single-stranded supramolecular helix of L,L-**AI**.

NMR spectra were generated to bring evidence for the C−I···S halogen bonding that bridges to form the supramolecular double helix in solution phase. The β-turn structures in L,L-**XA**s were confirmed by [1]H NMR traces in DMSO-$d_6$/$CD_3CN$ binary solvents of increasing content of the hydrogen bonding component DMSO-$d_6$, that the resonance of the thioureido $-NH_d$ proton (Fig. 3a) changed slightly upon increasing the percentage of DMSO-$d_6$ (Supplementary Figs. 35–37)[53]. In addition, the temperature coefficient of the chemical shift of $-NH_d$ ($\Delta\delta/\Delta T = -3.2$ ppb/°C) in L,L-**IA** is much smaller than those of $-NH_a$ (−9.3), $-NH_b$ (−9.3), and $-NH_c$ (−10.6), both supporting that the $-NH_d$ proton is intramolecularly hydrogen bonded to form the β-turn structures (Supplementary Figs. 38 and 39)[54]. Due to limited solubility of L,L-**XA**s in $CD_3CN$, [1]H NMRs were taken from saturated solutions in $CD_3CN$ on an 850 MHz NMR instrument. L,L-**IA** exhibited two sets of [1]H NMR resonances (Fig. 4a and Supplementary Fig. 40), especially the signals of the $-CH_3$ protons in the alanine residue ($H_h$ and $H_{h'}$). Referring to the [1]H NMR of control compounds L,L-**XA** (X ≠ I) that display only one set of [1]H NMR resonance as an indication of their monomer in *trans*-form, the stronger signal of L,L-**IA** is assigned to the monomer in the *trans*-form, while the weaker one to the helical oligomers consisting of L,L-**IA** in *cis*-form (Fig. 4a). This assignment is further supported by the weakened signal of $H_{h'}$ in $CD_3CN/D_2O$ mixtures compared to that in pure $CD_3CN$ (Supplementary Fig. 41), and by the concentration-dependent [1]H NMR spectra that the relative intensity of the signal corresponding to the oligomers declined upon diluting (Supplementary Fig. 42). [1]H NMR spectrum of L,L-**IA** in $CD_3CN$ using $CH_2Cl_2$ as an internal standard also supported the presence of large polymeric species (Supplementary Fig. 43), as concluded for the formation of polymeric structures of L,L-**IA** in $CH_3CN$ manifested by the CD spectra and DLS results (Fig. 2b and Supplementary Fig. 11). Moreover, DFT calculations showed that the resonance of the $-CH_3$ groups in the double helical oligomers of L,L-**IA** in *cis*-form (1.42 ppm) appeared at higher field than that of the L,L-**IA** monomer in *trans*-form (1.50 ppm, Fig. 4a and Supplementary Fig. 44), which is in agreement with the experimental assignment to the $H_{h'}$ of oligomers in the *cis*-form and to $H_h$ of the L,L-**IA** monomer in *trans*-form.

2D NOESY experiments next were performed to acquire direct evidence for the double helical structure in solution. Adjacent $H_f$ and $H_g$ at the I-substituted phenyl rings in L,L-**IA** showed obvious NOE signals in $CD_3CN$, while the contact between $H_e$ and $H_f$ can be attributed to the folded β-turn structure (Fig. 4b). These two peaks were also observed in DMSO-$d_6$ in which L,L-**IA** exists in its monomeric form, meaning the existence of the folded β-turn structure in DMSO-$d_6$ (Supplementary Fig. 45). Significantly, distinct NOE peaks seemingly between $H_e$ and $H_g$, of comparable intensity to the couplings of adjacent $H_f$-$H_g$, were

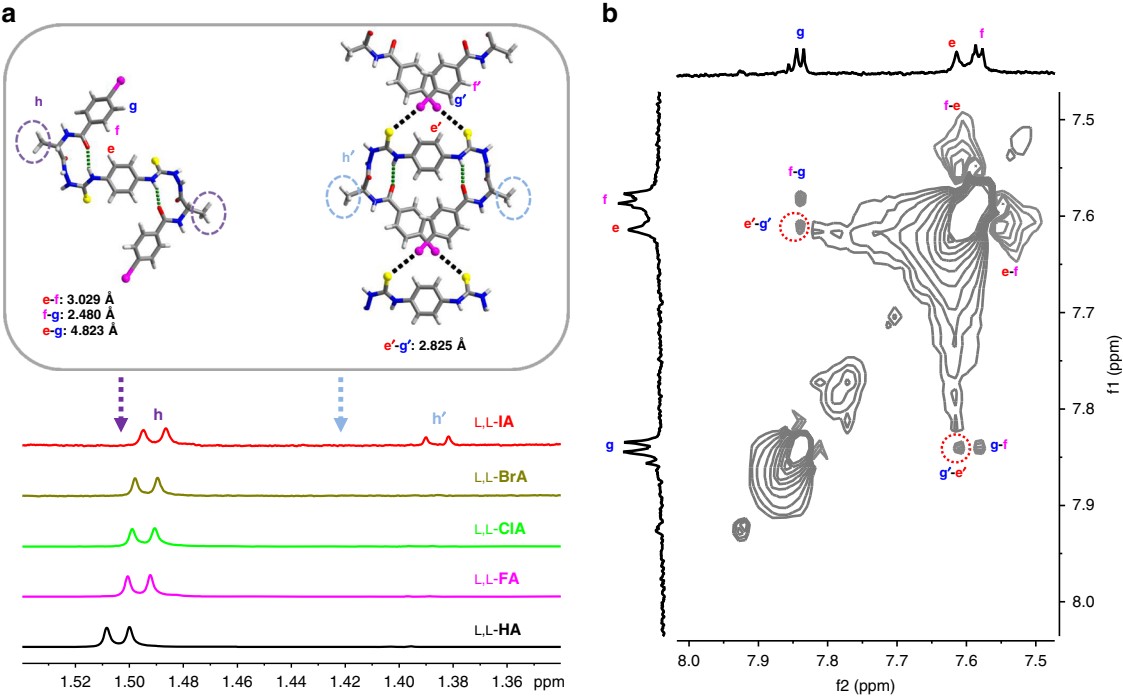

**Fig. 4** $^1$H NMR studies. **a** Partial 850 MHz $^1$H NMR spectra of $-CH_{3h}$ in L,L-**XA** in CD$_3$CN at 25 °C. The solutions were samples saturated. Dashed arrowed lines illustrate DFT calculated chemical shifts of the $-CH_3$ groups in the monomer of L,L-**IA** in the *trans*-form (1.50 ppm) and in the oligomers of L,L-**IA** in the *cis*-form (1.42 ppm) that support the assignments of the $^1$H NMR signals of $-CH_3$ groups. The structures with protons and distances labeled are also shown. Method for NMR calculation: DFT OPBE with the 6–311 + G(2d, p) basis set for C, H, O, N, and S atoms, and LANL2DZ for I atom. **b** Expanded 2D NOESY spectrum (850 MHz, 25 °C, mixing time 800 ms) of couplings between protons in phenyl rings in L,L-**IA** molecules in CD$_3$CN at 5.6 μM

observed for L,L-**IA** in CD$_3$CN (Fig. 4b). This seemingly H$_e$–H$_g$ coupling, however, was not observed in DMSO-$d_6$, despite with the similar intramolecular β-turn structure (Supplementary Fig. 45). This, together with the much longer H$_e$–H$_g$ distance of *ca*. 4.823 Å than the H$_f$–H$_g$ distance (*ca*. 2.480 Å, both from the calculated structure of *trans*-form L,L-**IA** in CH$_3$CN, Fig. 4a) and the dynamic nature of the halogen-bonded double helices, suggests that the NOE signals of the seemingly intramolecular H$_e$-H$_g$ coupling are actually those of the intermolecular coupling between H$_{e'}$ and H$_{g'}$, respectively, from two adjacent *cis*-form L,L-**IA** molecules in the double helices, in which H$_{e'}$ and H$_{g'}$ are brought by the crossed double C−I···S halogen bonds into close proximity of distance 2.825 Å (from the calculated oligomers of L,L-**IA** in *cis*-form in CH$_3$CN, Fig. 4a). These intermolecular H$_{e'}$–H$_{g'}$ couplings were also tested for the single-stranded helical structure. By checking the β-turn structured L,L-**IA**, we proposed a right-handed single-stranded helix formed from the *trans*-form L,L-**IA** molecules as driven by the head-to-tail C−I···π halogen bonding (Supplementary Fig. 46). However, in this case the H$_{e'}$ and H$_{g'}$ protons, respectively, from two adjacent molecules are too distant from each other by 8.687 Å that would not lead to intermolecular NOE peaks. The solution 1D and 2D NMR data therefore supported the supramolecular double helix structures of *cis*-form L,L-**IA** in the solution phase.

Note that for the two adjacent L,L-**IA** or D,D-**IA** molecules in *cis*-form, the I and S atoms are well oriented to allow the crossed double C−I···S halogen bonds (Fig. 2b) that support the supramolecular double helix. In this context, it was expected that similar double halogen bonds could not form between the *cis*-from L,L-**IA** and D,D-**IA** molecules since the orientations of I and S atoms are not matched, as well as the helical β-turn fragments (Fig. 5a). This means that a homochiral self-sorting[55–59] in the supramolecular structures formed from the enantiomeric mixtures of L,L-**IA** and D,D-**IA** may takes place.

CD spectra of the mixtures of L,L-/D,D-**IA** of varying *ee* were therefore recorded in CH$_3$CN (Supplementary Fig. 47). We found that the CD signal at 291 nm varied linearly with *ee* (Fig. 5b), a character reported in homochiral supramolecular systems[60,61]. SEM images of L,L-**IA**, D,D-**IA** and their racemate showed similar rod-like aggregates (Supplementary Fig. 48), again supporting the formation of homochiral assemblies. DLS data exhibited similar polymeric size for L,L-**IA**, D,D-**IA** and their racemate in solution (Supplementary Fig. 49) and the absorption spectrum of the racemate was the same as that of the mathematical addition of the absorption spectra of L,L-**IA** and D,D-**IA** (Supplementary Fig. 50), both agreeing with the homochiral self-sorting in solution that L,L-**IA** and D,D-**IA** are separated. This was further supported by the 850 MHz $^1$H NMR spectra that the signals of H$_h$ and H$_{h'}$ of the racemate, especially that of H$_{h'}$ that is assigned to the helical oligomers in *cis*-form, are of the same chemical shifts as those of enantiomer L,L-**IA** or D,D-**IA** (Fig. 5c). We therefore concluded that the homochiral self-sorting occurred that the *P*-double and *M*-double helices were formed separately in solution from the enantiomer mixtures of L,L-**IA** and D,D-**IA**.

## Discussion

In summary, we developed the *N*-(*p*-iodobenzoyl)alanine based bilateral amidothiourea framework that allows, in the otherwise disfavored *cis*-form, the intermolecular double and crossed C−I···S halogen bonds that support the single-handed supramolecular double helix, in the solid state and in the extremely dilute solution as well. The *N*-(*p*-iodobenzoyl)alanine based amidothiourea moiety holds a helical β-turn structure, while the bilateral *N*-amidothiourea of homochiral configuration, L,L-**IA** or D,D-**IA**, allows the intermolecular double and crossed C−I···S halogen bonds so that the helical β-turn structures are linked into two helical strands of the same helicity, which are intertwined by

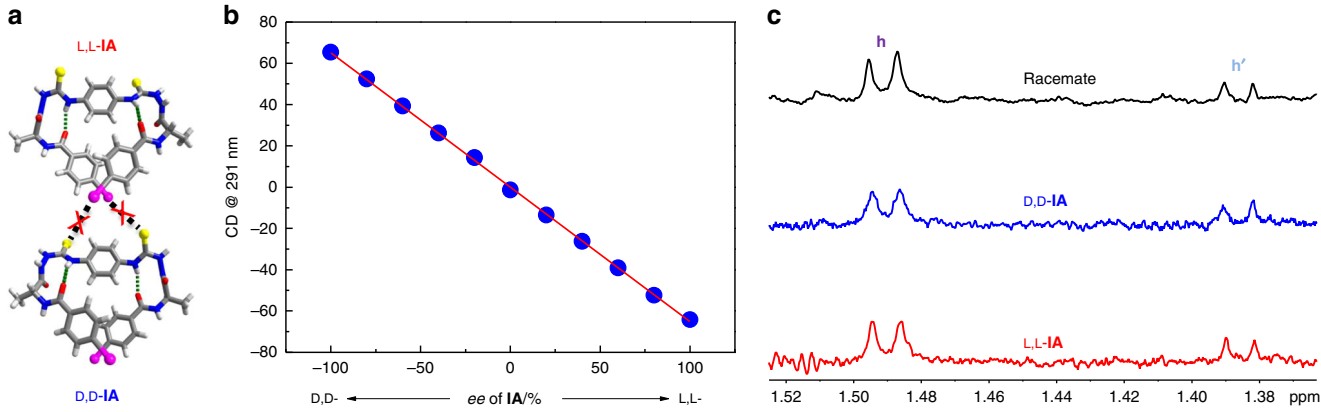

**Fig. 5** Study of homochiral self-sorting. **a** Illustration that the crossed C−I···S halogen bonds between one L,L-**IA** and one D,D-**IA** molecules are unlikely because of the mismatched orientation of I-atom and S-atom. **b** Plots of CD signal at 291 nm of **IA** in CH$_3$CN against ee. [L,L-**IA**] + [D,D-**IA**] = 5 μM. **c** Partial 850 MHz $^1$H NMR spectra of –CH$_{3h}$ in L,L-**IA**, D,D-**IA** and their racemate in CD$_3$CN at 25 °C. The solutions were sample saturated

the central *p*-phenylenediamine covalent linkage. Homochirality plays a decisive role in the formation of the double helix, as the helical structures did not exist when the heterochiral L,D-**IA** was employed. This was supported by the observation of the homo-chiral self-sorting in the formed supramolecular double helix from the enantiomeric mixtures of L,L-**IA** and D,D-**IA** in solu-tions, a character that is attributed to the high stability of the supramolecular double helix manifested by the formation of double helix at low concentration of the micromolar level, i.e., 5.6 μM from 2D NOESY experiments, and a high thermal sta-bility up to 75 °C in CH$_3$CN with almost unchanged CD spectra. These differ quite significantly from those of the single stranded helix built via the head-to-tail like C−I···π halogen bonding[22], which showed for example a chiral amplification characterized by the "S"-shaped dependence of the CD signal versus ee. Our success of building a supramolecular double helix in dilute solution phase provides a unique approach towards the artificial double helices from synthetic small molecules.

## Methods

**Synthesis and characterization**. Synthetic procedures for L,L-**IA**, D,D-**IA**, L,D-**IA** and control compounds are given in Supplementary Figs. 1 and 2 in the Supple-mentary Information. Experimental and characterization details can be found in the Supplementary Methods.

**General methods**. Absorption spectra were recorded on a Thermo Scientific Evolution 300 UV/Vis spectrophotometer. CD spectra were recorded with a JASCO J-810 circular dichroism spectrometer. Infrared spectra were carried out using a Nicolet AVATAR FT-IR330 spectrometer. Raman spectra were recorded with a Jobin-Yvon Horiba Xplora confocal Raman system. $^1$H NMR, $^{13}$C NMR and 2D NOESY spectra were obtained on Bruker AV500 MHz or AV850 MHz spec-trometer in acetonitrile-D$_3$ (CD$_3$CN), dimethyl sulfoxide-D$_6$ (DMSO-$d_6$) or mixed solvents. High-resolution mass spectra (HR-MS) were obtained on a Bruker En Apex ultra 7.0 FT-MS. DLS were collected with a Malvern Zetasizer Nano-ZS90. SEM experiments were conducted by using a Hitachi S-4800 scanning electron microscope. STM experiments were performed using a Nanoscope E STM system (Veeco, Japan) with a mechanically cut Pt-Ir tip. All calculations were carried out using Gaussian 09. X-ray crystallography data of compounds L,L-**IA**, D,D-**IA**, and L,D-**IA** were collected on an Agilent SuperNova Dual system (CuKα, λ = 1.54184 Å) at 293 K. Absorption corrections were applied by using the program CrysAlis (multi-scan). The structure was solved by direct methods, and non-hydrogen atoms were refined anisotropically by least-squares on $F^2$ using the SHELXTL program. For L,L-**IA** and D,D-**IA**, the diffuse electron densities resulting from the residual solvent molecules were removed from the data set using the SQUEEZE routine of PLATON.

**Reporting summary**. Further information on research design is available in the Nature Research Reporting Summary linked to this article.

## Data availability

The X-ray crystallographic coordinates for the structures reported in this article have been deposited at the Cambridge Crystallographic Data Centre (CCDC), under deposition number CCDC 1584955 (D,D-**IA**), 1584956 (L,D-**IA**), 1584957 (L,L-**IA**). These data can be obtained free of charge from The Cambridge Crystallographic Data Centre via www.ccdc.cam.ac.uk/data_request/cif. The data that support the findings of this study are available within the paper and its Supplementary Information files or from the corresponding author upon reasonable request.

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

## Acknowledgements

This work was supported by the NSF of China (Grants 21435003, 91427304, 91856118, 21521004, 21820102006, and J1310024), and the Program for Changjiang Scholars and Innovative Research Team in University, administrated by the MOE of China (Grant IRT13036), and Xiamen University President Foundation (20720170088). We thank Xiaoyu Cao, Zhuoru Li, and Philip A. Gale for discussions and Hongwei Yao and Liubin Feng for help on the NMR measurements, Chuanfan Ding for ESI-HRMS measurements, Weitai Wu and Xuezhen Lin for helpful measurements on DLS, Ziang Nan for help on the analysis of crystallographic data, and Haisheng Su and Qinqing Zhao for helpful measurements on STM.

## Author contributions

X.Y. and Y.J. designed the experiments and analyzed the data. X.Y. performed most of the experiments. J.C. analyzed the crystal data. K.Z., X.L. and Z.Z. contributed to the SEM amd NMR experiments. Z.L. contributed to the discussions. A.W., W.L. and Y.M. provided suggestions on the DFT calcualtions. X.Y. wrote the paper with the contributions from all authors.

## Additional information

**Competing interests:** The authors declare no competing interests.

