## [Peer Review File · Nature Communications]

Reviewers' Comments:

Reviewer #1:

Remarks to the Author:

It is an interesting manuscript describing the formation of supramolecular double helical structure through C-I...S halogen bonding. Double helical structure of the I-substituted bilateral N-amidothiurea crystal was characterized by single crystal x-ray analysis together with simulation, but the evidence of double helical structure formation in solution is not conclusive. The positive Cotton effect of L,L-IA at long wavelength 342 nm and also similar CD spectra of L,L-IA and D,D-IA in CH₃CN solutions to those in the solid state indicated the P-helical structures in CH₃CN solution, but it is not a clear evidence for the formation of double helical structure in solution. Stability at much lower concentration but in the similar temperature range also does not guarantee the double helical structure. They, IA and AI, are not a same molecule. SEM images of air-dried samples of L,L-IA and D,D-IA in CH₃CN showed ordered rod-shaped aggregates, not screwed shape. Solution x-ray diffraction may prove the formation of double helical structure of I-substituted bilateral N-amidothiureas.

Reviewer #2:

Remarks to the Author:

This paper reports a pair of artificial double helices by self-assembly of small organic molecules via strong inter-molecular non-covalent halogen bonds. A bilateral N-(p-iodobenzoyl)alanine based amidothiurea was designed and synthesized for this purpose which in its folded cis-form allows double and crossed C-I...S halogen bonds that is favorable to the formation of double helix. The double helix was found to form in solid state and also in dilute solution. The construction of double helix using a small organic molecule via halogen bonds, especially in dilute solution, is new and could be of interest to the readers of the community. In general, the manuscript is well written and publication may be considered after addressing the following questions.

- 1) The authors claimed that the double helix forms in CH₃CN solution at an extremely low concentration of 0.08 μ M, mainly supported by the concentration dependent CD spectra measurements (Fig. 3d and Supplementary Fig. 12). However, NMR spectra do not fully agree with this conclusion. As demonstrated, the signals of the -CH₃ protons in the alanine residue show two sets of ¹H NMR resonance, the stronger one assigned to the monomer in the trans-form and the weaker one to the helical oligomers consisting of L,L-IA in cis-form. According to Supplementary Fig. 31, the weaker signals disappear at a concentration of 1.12 μ M, which is much higher than the value (0.08 μ M) obtained from the CD spectra. The NMR result suggested that the double helix may not be formed below 1.12 μ M. Please explain.
- 2) Dynamic light scattering (DLS) was measured to support the formation of helical aggregates. As the size of the aggregates can vary with the concentration and temperature, it is highly demanded to measure the concentration and temperature dependent DLS, and derive the corresponding relationships.
- 3) Homochiral self-sorting occurred in the supramolecular system of enantiomeric mixtures of L,L-IA and D,D-IA, and formation of conglomerates was suggested based on the CD measurements. However, further evidence is needed to support this conclusion. For example, single crystal structural determination on the randomly selected single crystals of the conglomerate should be performed.

Reviewer #3:

Remarks to the Author:

This contribution details very nicely the formation and structure of supramolecular helical structures in the solid state and in solution. The helices are based on halogen bonding and exhibit relative high thermal stability in solution – which is rare. It is very challenging to design such supramolecular structures. The homochiral self-sorting is another interesting observation. The new materials are well-characterized (high-resolution NMR, X-ray analysis, CD spectroscopy). Publication in Nature Communications is recommended, however in revision I would like that the authors provide additional spectroscopic information (infrared spectroscopy for example) to provide direct evidence for halogen bonding in solution. Binding constants in solution could be added to the manuscript. Note that figure 3 of the ESI is more clear than the figure provided in the main text.

Reviewer #4:

Remarks to the Author:

As requested, I comment on the crystallographic part only:

- 1: As it is good practice, the authors should include the .res and .hkl-files in the provided cif-files. This can be done automatically when using Olex2 with an up-to-date version of SHELXL
2. The sum-formulae used for the X-ray diffraction experiments should be the chemically correct formulae and not just the sum of the atoms used for the refinement. So, all hydrogen atoms of methyl-groups and water molecules should be included.
3. H-atoms of the methy-groups of the DMF and the dimethylcarbamate (is this correct or is there a mistake in the description of the respective solvent/counterion in the triclinic structure?) should be included in the refinement (for example with AFIX 137)
4. The standard uncertainties for the I...S halogen bonds should be given and then determined, if they deviate significantly (3σ) from the sum of the Van-der-Waals radii. A discussion with a comparison of the I...S-distances found in other compounds should be given in order to support the claim for I...S halogen bonds.
5. Standard uncertainties should be given for all values in Supplementary tables 2 and 3.
6. The $wR2$ values are much larger than about twice the $R1$. Since there are no input files for the refinement provided, I cannot check different possible causes.

Point-by-point response to the reviewers' comments

Reviewer #1 (Remarks to the Author):

It is an interesting manuscript describing the formation of supramolecular double helical structure through C-I...S halogen bonding. Double helical structure of the I-substituted bilateral *N*-amidothiourea crystal was characterized by single crystal x-ray analysis together with simulation, but the evidence of double helical structure formation in solution is not conclusive. The positive Cotton effect of L,L-**IA** at long wavelength 342 nm and also similar CD spectra of L,L-**IA** and D,D-**IA** in CH₃CN solutions to those in the solid state indicated the *P*-helical structures in CH₃CN solution, but it is not a clear evidence for the formation of double helical structure in solution. Stability at much lower concentration but in the similar temperature range also does not guarantee the double helical structure. They, **IA** and **AI**, are not a same molecule. SEM images of air-dried samples of L,L-**IA** and D,D-**IA** in CH₃CN showed ordered rod-shaped aggregates, not screwed shape. Solution X-ray diffraction may prove the formation of double helical structure of I-substituted bilateral *N*-amidothioureas.

Response: Following the reviewer's suggestion, we carried out a series of new experiments, in an attempt to bring in more evidence for the double helical structure in solution phase, *i.e.* solution X-ray diffraction, IR and Raman spectroscopy. Due to the very low concentration (5 μM) that we could make because of the limited solubility, we were unable to obtain credible signals from those highly dilute solutions.

We actually showed that, in addition to the similarity of the CD spectra in solution phase to those in the solid phase (Figure 3c) that supports the double helical structure in solution, high resolution ¹H NMR spectra also helped to characterize the solution phase double helical structure as well, in particular when coupled with the DFT calculations. The observed ¹H NMR signals of the oligomers of *cis*-form L,L-**IA** (Figure 4c and Supplementary Figure 32) agreed with those calculated from DFT computations using double helical structural parameters obtained in the crystal structure (Supplementary Figure 33), supporting the double helical structure of the *cis*-form L,L-**IA** in solution.

In addition, the following facts also helped the conclusion of the double helical structure in solution. (i) The CD spectral profile of L,L-**IA** in solution (Figure 3b) is similar to that of the natural DNA, with the first positive, the second negative and the third positive Cotton effects that are indicative of the *P*-helicity, in agreement with the *P*-double helix identified for L,L-**IA** in the crystals, and therefore concluded in CH₃CN solution as well. We have highlighted this discussion in the main text at Page 7 of the revised manuscript. (ii) Concentration and temperature dependent CD spectral data (Supplementary Figures 11-14) suggested the stable helical structures in solution at low concentration and at high temperature, which were further supported by the corresponding concentration and temperature dependent DLS data (Supplementary Figures 16 and 17. Also please see Figures R1 and R2 later in this Response). These can be attributed to the crossed double halogen bonds since the crossed structure is a highly stable geometry, supporting the formation of double helix in solution with high stability.

The formed double helix is 1D superstructure, while the SEM images of air-dried samples showed 3D structures, we observed in the SEM thus the ordered rod-shaped aggregates. This holds similarity to the 3D supramolecular architectures shown in Supplementary Figures 3 and 4 in the crystal structure of L,L-**IA**.

Reviewer #2 (Remarks to the Author):

This paper reports a pair of artificial double helices by self-assembly of small organic molecules via strong inter-molecular non-covalent halogen bonds. A bilateral *N*-(*p*-iodobenzoyl)alanine based amidothiourea was designed and synthesized for this purpose which in its folded *cis*-form allows double and crossed C-I \cdots S halogen bonds that is favorable to the formation of double helix. The double helix was found to form in solid state and also in dilute solution. The construction of double helix using a small organic molecule via halogen bonds, especially in dilute solution, is new and could be of interest to the readers of the community. In general, the manuscript is well written and publication may be considered after addressing the following questions.

Response: We thank very much the reviewer for his/her positive comments. We also appreciate the constructive comments which we respond in the following.

1) The authors claimed that the double helix forms in CH₃CN solution at an extremely low concentration of 0.08 μ M, mainly supported by the concentration dependent CD spectra measurements (Fig. 3d and Supplementary Fig. 12). However, NMR spectra do not fully agree with this conclusion. As demonstrated, the signals of the -CH₃ protons in the alanine residue show two sets of ¹H NMR resonance, the stronger one assigned to the monomer in the *trans*-form and the weaker one to the helical oligomers consisting of L,L-**IA** in *cis*-form. According to Supplementary Fig. 31, the weaker signals disappear at a concentration of 1.12 μ M, which is much higher than the value (0.08 μ M) obtained from the CD spectra. The NMR result suggested that the double helix may not be formed below 1.12 μ M. Please explain.

Response: Thanks for pointing to us this point. Compared to the CD spectrometry, the ¹H NMR technique is of lower sensitivity, especially to the oligomeric or polymeric species, even the high resolution (850 MHz) NMR instrument we employed here could not help too much. When the sample solutions for NMR measurements were very dilute, *e.g.* 1.12 μ M, in which the oligomers only stand for a small fraction, their NMR signals could not be detected. Actually NMR signals of the helical oligomers could not be observed when the concentration was below 2.24 μ M (Supplementary Figure 32), despite the fact that the CD spectrometry has shown the occurrence of the double helical structure at a much lower concentration of 0.08 μ M (Figure 3d). To avoid the possible confusion, we have added an explanation in the legend of the Supplementary Figure 32 in the revised ESI. "At lower concentration, *i.e.* 1.12 μ M, due to the sensitivity limit of the NMR technique, in particular to the oligomeric species, the signals for the helical oligomers of the *cis*-form L,L-**IA** (h') could not be observed".

2) Dynamic light scattering (DLS) was measured to support the formation of helical aggregates. As the size of the aggregates can vary with the concentration and temperature, it is highly demanded to measure the concentration and temperature dependent DLS, and derive the corresponding relationships.

Response: Thanks for this suggestion. We have accordingly carried out the concentration and temperature dependent DLS measurements.

We found from the DLS data (Fig. R1) that, while varying solution concentration from 0.5 to 5 μM , the “size” of the species existing in solution varied little. This suggests that the existence of the supramolecular polymeric species of L,L-IA in CH_3CN even at a very low concentration and the increase in concentration mainly results in the increase in the number, not the length, of the polymeric species. Measurements at concentrations lower than 0.5 μM were not possible due to the limit of the sensitivity.

Figure R1 | Concentration-dependent (0.5 to 5 μM) hydrodynamic diameters of L,L-IA in CH_3CN measured by dynamic light scattering at 25 $^\circ\text{C}$. Measurements at concentrations lower than 0.5 μM were not possible due to the limit of the DLS sensitivity.

Figure R2 | Temperature-dependent (25 to 75 °C) hydrodynamic diameters of *L,L*-**IA** in CH_3CN measured by dynamic light scattering. Inset is the mean size versus temperature. [*L,L*-**IA**] = 5 μM .

In the temperature dependent DLS data, we observed decreased size of the polymeric species when the solution was heated from 25 to 75 °C (Fig. R2). The observation that the large size species of *ca.*100 nm existed at 75 °C, again indicates that the superstructures are stable at high temperature. Since the CD signal also did not drop at high temperature (Supplementary Figure 12), we concluded that heating of the solution led to a decrease in the length, while an increase in the number of the supramolecular polymeric species in the solution. These observations, made from the concentration and temperature dependent DLS measurements, demonstrated that the supramolecular structures in CH_3CN exist at extremely low concentration and remain at high temperature, in agreement with the conclusions made from the concentration and temperature dependent CD spectral data.

We have added these DLS results in the revised ESI, as Supplementary Figures 16 and 17, and the relevant discussion was added and highlighted in the main text at Page 8 of the revised manuscript.

3) Homochiral self-sorting occurred in the supramolecular system of enantiomeric mixtures of *L,L*-**IA** and *D,D*-**IA**, and formation of conglomerates was suggested based on the CD measurements. However, further evidence is needed to support this conclusion. For example, single crystal structural determination on the randomly selected single crystals of the conglomerate should be performed.

Response: In the revision, we have provided further evidences to support the homochiral self-sorting in solution as driven by halogen bonds. First, *L,L*-**IA**, *D,D*-**IA** and their racemate exhibited similar polymeric size in solution, according to the DLS measurements (Fig. R3). Second, the absorption spectrum of the racemate was the same as the mathematical addition of the absorption spectra of *L,L*-**IA** and *D,D*-**IA** (Fig. R4). Moreover, in the 850 MHz ^1H NMR spectra, the signals of H_h and H_h' for the racemate, especially H_h' , that is assigned to the helical oligomers in *cis*-form,

are of the same chemical shifts to those of enantiomer L,L-**IA** or D,D-**IA**, (Fig. R5). These results are in accordance with the homochiral self-sorting in solution that L,L-**IA** and D,D-**IA** are separated.

We have added these results as Supplementary Figures 36-38 in the revised ESI, and the relevant discussion was highlighted in the main text at Pages 10 and 11.

Figure R3 | Hydrodynamic diameters of L,L-**IA** (a), D,D-**IA** (b) and their racemate (c) in CH₃CN measured by dynamic light scattering at 25 °C.

Figure R4 | Absorption spectra of L,L-**IA**, D,D-**IA** and their racemate in CH₃CN at 25 °C. The black line is calculated by the addition of L,L-**IA** (red line) and D,D-**IA** (blue line), practically the same as the experimental spectrum of the racemate (green line).

Figure R5 | Partial 850 MHz ^1H NMR spectra of $-\text{CH}_{3\text{h}}$ in L,L-**IA**, D,D-**IA** and their racemate in CD_3CN at 25 °C. The solutions were sample saturated.

The homochiral self-sorting in CH_3CN solution results from the fact that the orientations of I and S atoms are not matched between the *cis*-form L,L-**IA** and D,D-**IA** molecules so to afford double crossed halogen bonds (Figure 5a). In the solid state, in addition to the halogen bonding to lead to the 1D double helical structure, intermolecular hydrogen bonds also occur to drive the formation of 3D supramolecular architectures as shown in Supplementary Figures 3 and 4 in the crystal structure of L,L-**IA**. Since we have been unable to obtain the crystals of the racemate by using the same routine for L,L-**IA** and D,D-**IA** and after months of trials of many other attempts of varying solvent compositions and rates of crystallization, we at this stage could not assume the formation of conglomerates in the solid racemate due to the uncertainty of hydrogen bonding. To avoid the possible misleading, in the revised manuscript we emphasized the homochiral self-sorting in the solution phase as driven by halogen bonding and removed the statement of “the formation of conglomerates”.

Reviewer #3 (Remarks to the Author):

This contribution details very nicely the formation and structure of supramolecular helical structures in the solid state and in solution. The helices are based on halogen bonding and exhibit relative high thermal stability in solution – which is rare. It is very challenging to design such supramolecular structures. The homochiral self-sorting is another interesting observation. The new materials are well-characterized (high-resolution NMR, X-ray analysis, CD spectroscopy). Publication in Nature Communications is recommended, however in revision I would like that the authors provide additional spectroscopic information (infrared spectroscopy for example) to provide direct evidence for halogen bonding in solution. Binding constants in solution could be added to the manuscript. Note that figure 3 of the ESI is more clear than the figure provided in the main text.

Response: We are delighted that the reviewer has noted the originality and importance of our work and recommended for publication after minor revision.

To provide direct evidence for halogen bonding in solution, we tried solution X-ray diffraction, IR and Raman spectroscopy, but could not obtain signals due to the very low concentration (5 μM) that is out of the limit of detection of these measurements.

We have added the binding constant (equilibrium constant, K_e) in solution, $1.25 \times 10^7 \text{ M}^{-1}$ at 25 °C, according to the critical aggregation concentration that has been suggested to be close to K_e^{-1} by the reference: Smulders, M. M. J. *et al.* How to distinguish isodesmic from cooperative supramolecular polymerisation. *Chem. – Eur. J.* **16**, 362-367 (2010). This is now highlighted at Page 8 in the main text.

By combining Figure 2 in the main text and original Supplementary Figure 3 in the ESI, we provided a clearer figure (Figure 2 at Page 5) in the revised manuscript, whereas the original Supplementary Figure 3 in ESI was removed.

Reviewer #4 (Remarks to the Author):

As requested, I comment on the crystallographic part only:

1. As it is good practice, the authors should include the .res and .hkl-files in the provided cif-files. This can be done automatically when using Olex2 with an up-to-date version of SHELXL

Response: We have now included the .res and .hkl-files in the provided cif-files.

2. The sum-formulae used for the X-ray diffraction experiments should be the chemically correct formulae and not just the sum of the atoms used for the refinement. So, all hydrogen atoms of methyl-groups and water molecules should be included.

Response: We have now provided the chemically correct formulae, and all hydrogen atoms of solvent have been included.

3. H-atoms of the methyl-groups of the DMF and the dimethylcarbamate (is this correct or is there a mistake in the description of the respective solvent/counterion in the triclinic structure?) should be included in the refinement (for example with AFIX 137)

Response: All H-atoms have now been included in the refinement. For L,L-**IA** and D,D-**IA**, the diffuse electron densities resulting from the residual solvent molecules were removed from the data set using the SQUEEZE routine of PLATON. We have added this description at the end of “General methods”.

4. The standard uncertainties for the I \cdots S halogen bonds should be given and then determined, if they deviate significantly (3sigma) from the sum of the Van-der-Waals radii. A discussion with a comparison of the I \cdots S-distances found in other compounds should be given in order to support the claim for I \cdots S halogen bonds.

Response: The standard uncertainties for I \cdots S halogen bonds have been given in the revision, and we have added the discussion of the I \cdots S-distances in the revised manuscript at Page 4. In addition to the bond length (3.764 Å) that is

shorter than the sum of the van der Waals radii of I and S atoms (3.780 Å), the calculated binding energies again support the claim for I...S halogen bonds.

5. Standard uncertainties should be given for all values in Supplementary tables 2 and 3.

Response: We have added standard uncertainties for all values in Supplementary Tables 2 and 3.

6. The wR2 values are much larger than about twice the R1. Since there are no input files for the refinement provided, I cannot check different possible causes.

Response: After refinement, these values have been decreased, and we have also provided input files for the refinement to enable the checking of the possible causes.

We thank the reviewer for the careful checking of our crystallographic part, and we do hope that the revised crystal data are now better suited for publication.

Reviewers' Comments:

Reviewer #1:

Remarks to the Author:

In the revised manuscript, authors do not provide any further evidence for the formation of double helical structures in solution. CD spectra are clear evidence for the formation of chiral helical structure, but not for double helical structures. As described in the response to the reviewers' comments, they stated that "high resolution ^1H NMR spectra also helped to characterize the solution phase double helical structure as well, in particular when coupled with the DFT calculations. The observed ^1H NMR signals of the oligomers of cis-form L,L-IA (Figure 4c and Supplementary Figure 32) agreed with those calculated from DFT computations using double helical structural parameters obtained in the crystal structure (Supplementary Figure 33), supporting the double helical structure of the cis-form L,L-IA in solution". However, in the response to the comment raised by reviewer 2, they also wrote that "When the sample solutions for NMR measurements were very dilute, e.g. $1.12\ \mu\text{M}$, in which the oligomers only stand for a small fraction, their NMR signals could not be detected. Actually NMR signals of the helical oligomers could not be observed when the concentration was below $2.24\ \mu\text{M}$ (Supplementary Figure 32)". It should be cleared before the publication of this manuscript.

Reviewer #2:

Remarks to the Author:

I am satisfied with the authors' response concerned with my questions. Therefore, publication of this work is recommended.

Reviewer #3:

Remarks to the Author:

The control experiments indicate that the halide plays an important role in formation the observed supramolecular structures in solution. However, the rather long "halogen-bond" distances and the angles in the solid state makes me believe that direct spectroscopic measurements are needed to support the claims made by the authors in this manuscript.

Reviewer #4:

Remarks to the Author:

Concerning the crystallography part:

most of the issues with the crystal structures analyses have been resolved. However, there are still some problems with the disordered DMF-molecules:

The authors described one of the DMF molecules with one fully occupied position (see attached figure a)). If the DMF molecules are described as a superposition of two rigid-groups with almost equal occupancies (see attached figure b)), the R1-value drops about 1% and the alerts in the checkcif-routine related to wrongly assigned H-atoms disappear.

Minor issue: the table "Supplementary Table 2" contains a lot of standard uncertainties, which seem to be way too large. Please check.

General remarks:

I have some severe concerns, regarding the main claims of the manuscript:

Despite the nice illustrations (Fig.1) showing the relation to the DNA double helix, one should not forget that the similarity is of purely graphical nature and it does not reflect similarities with the bonding situation in DNA at all. In fact, in DNA the situation and therefore the energetics is completely different, which means that one cannot simply expect that both systems behave in the same manner, neither in solid state nor in solution.

The claim of a significant contribution of halogen bonds in the crystal is not convincing in my eyes. The distances are with 3.76 Å as large as the sum of the van der Waals radii (3.78 Å) of I and S and therefore MUCH (!) larger than in "classical" systems where halogen bonds have a significant contribution (3.2-3.4 Å, see page 4 line 106-109). There are not enough details given for the DFT calculations, to judge, if even the (small) contribution of 12 KJ/mol for the halogen bonds is reliable, since there is disorder in the structure which has to be resolved and correctly be taken into account for DFT calculations.

Especially in solution, I strongly doubt, that the weak halogen bond contribution alone is large enough to trigger the aggregation of hundreds of molecules together, forming chains of 255 nm in size. We do not have a covalently bonded backbone or the concerted formation of many hydrogen bonds leading to the formation of the double helix like we see it in DNA, since in IA each monomer is connected to its neighbors only by 2 weak halogen bonds. Accordingly, here no energetic gain can be obtained from the concerted formation of a special arrangement, like a double helix. This may also explain, why the authors do not see oligomers in the NMR experiments (see SI).

I see, that the authors have carefully performed many experiments. Still, for the above reasons I cannot recommend a publication of the manuscript in the present form, because I cannot see enough evidence for the main claims, as there are:

- 1) a significant contribution of halogen bonds to the packing in the crystal and
- 2) the formation of strands composed of halogen bonded monomers in solution.

a)

b)

Point-by-point response to the reviewers' comments

Reviewer #1 (Remarks to the Author):

In the revised manuscript, authors do not provide any further evidence for the formation of double helical structures in solution. CD spectra are clear evidence for the formation of chiral helical structure, but not for double helical structures. As described in the response to the reviewers' comments, they stated that "high resolution ^1H NMR spectra also helped to characterize the solution phase double helical structure as well, in particular when coupled with the DFT calculations. The observed ^1H NMR signals of the oligomers of *cis*-form L,L-**IA** (Figure 4c and Supplementary Figure 32) agreed with those calculated from DFT computations using double helical structural parameters obtained in the crystal structure (Supplementary Figure 33), supporting the double helical structure of the *cis*-form L,L-**IA** in solution". However, in the response to the comment raised by reviewer 2, they also wrote that "When the sample solutions for NMR measurements were very dilute, *e.g.* 1.12 μM , in which the oligomers only stand for a small fraction, their NMR signals could not be detected. Actually NMR signals of the helical oligomers could not be observed when the concentration was below 2.24 μM (Supplementary Figure 32)". It should be cleared before the publication of this manuscript.

Response: We are happy that in this newly revised version we are able to provide new evidence for the double helical structure of L,L-**IA** in the solution phase, from calculated CD spectra in the solutions, competition experiments by halogen anions at higher concentrations and 2D NOESY. We also measured the Raman and IR spectra of solid samples to bring further evidence to support the C-I \cdots S halogen bonding in the solid state.

In our last reply to the comments of the Reviewer, our statement "high resolution ^1H NMR spectra also helped to characterize the solution phase double helical structure as well, in particular when coupled with the DFT calculations. The observed ^1H NMR signals of the oligomers of *cis*-form L,L-**IA** (Figure 4c and Supplementary Figure 32) agreed with those calculated from DFT computations using double helical structural parameters obtained in the crystal structure (Supplementary Figure 33), supporting the double helical structure of the *cis*-form L,L-**IA** in solution" should have been better described as "the concentration dependent high resolution ^1H NMR...". What we actually found was that, with increasing concentration of L,L-**IA**, we were able to observe NMR signals at higher concentration, *i.e.* 2.24 μM , of the formed oligomers of L,L-**IA**, for example the NMR signal of the $-\text{CH}_3$ protons in the oligomers at 1.39 ppm while that in the monomeric form at 1.49 ppm (Figures 4a and 5c in the revised manuscript). DFT calculations using double helical structural parameters from the crystal structure, led to values of 1.42 ppm and 1.50 ppm respectively, agreeing with the experimental data. At lower concentration of 1.12 μM , however, NMR could not show signals that can be assigned to the oligomers because of the limit of the sensitivity (Supplementary Figure 39), despite the fact that the concentration dependent CD spectra (Supplementary Figure 15 and Fig. 3d) and DLS data (Supplementary Figure 20) showed the formation of helical structures already at that concentration.

(i) In the original manuscript, we reported that the CD spectrum of L,L-**IA** in CH₃CN solution is similar to that in the solid state in which the L,L-**IA** was shown by the crystal structure to exist in double helical structure (Fig. 3c), from which we assumed that L,L-**IA** in CH₃CN solution may take the double helical structure. It would better be concluded that this similarity may suggest that it exists in helical structure in the solution.

We have now performed theoretical calculations on the CD spectrum of L,L-**IA** in double helical structure in solution, hoping to compare to the experimental CD spectrum to support the double helical structure in solution. While the calculated CD spectra of *trans*-L,L-**IA** and *cis*-L,L-**IA** in monomer form in CH₃CN solution are very different from the experimentally observed CD spectrum of L,L-**IA** in CH₃CN (Fig. R1), the calculated CD spectra of the crossed double halogen-bonded dimer and trimer of *cis*-L,L-**IA** are similar to the experimental CD spectrum, featuring the first positive, the second negative and the third positive Cotton effects that are indicative of the *P*-helicity.

Fig. R1 | The experimental and calculated CD spectra of monomer and oligomers of L,L-**IA** in CH₃CN solutions. The structures for the calculations are shown in the right side. Method: TDDFT wb97XD with the 6-31G* basis set for C, H, O, N and S atoms, LANL2DZ for I atoms.

Therefore, the observed similarity of the CD spectrum of L,L-**IA** in CH₃CN solution to that in the solid state of the double helical structure would now be more strongly to support the double helical structure in solution phase.

We have added these DFT calculation results in the revised ESI in Supplementary Figure 13 and the relevant discussions were added and highlighted in the main text on page 8 of the revised manuscript.

(ii) In the original manuscript, we described that the CD spectra of L,L-**IA** in CH₃CN were not affected by 5.0 eq halogen anions Cl⁻, Br⁻ and I⁻ (Supplementary Figs. 22-25), suggesting a high stability of the supramolecular double helix in CH₃CN. We show now, in the revised manuscript, that the introduction of the halogen anions at higher concentration of up to 1000 eq results in a gradual dropping of the values of the CD signals of the supramolecular

helical structures (Fig. R2). This means the damaging of the double helical structure by halogen anions at much higher concentration. The order of the ability of damaging is $\Gamma^- > \text{Br}^- > \text{Cl}^-$ (Fig. R3), opposite to the order of the hydrogen bonding ability of these anions with the thiourea moiety (the anion binding receptor) in L,L-IA, but it is consistent with the halogen-bonding ability of the halogen anions. This also supports the role of the halogen bonding in supporting the supramolecular helical structures of L,L-IA in CH_3CN solution.

We have added these spectral results in the revised ESI as Supplementary Figures 26 and 27 and the relevant discussions were added and highlighted in the main text on page 9 of the revised manuscript.

Fig. R2 | Time-dependent CD spectra of L,L-IA in CH_3CN in the presence of 1000 eq I^- (a), Br^- (b) and Cl^- (c). $[\text{L,L-IA}] = 5 \mu\text{M}$, $[\text{I}^-] = [\text{Br}^-] = [\text{Cl}^-] = 5000 \mu\text{M}$. I^- , Br^- and Cl^- exist as their $(n\text{-Bu})_4\text{N}^+$ salt.

Fig. R3 | Time profiles of CD signals at 291 nm of L,L-IA in CH_3CN in the presence of 1000 eq I^- , Br^- and Cl^- . $[\text{L,L-IA}] = 5 \mu\text{M}$, $[\text{I}^-] = [\text{Br}^-] = [\text{Cl}^-] = 5000 \mu\text{M}$. I^- , Br^- and Cl^- exist as the $(n\text{-Bu})_4\text{N}^+$ salt.

(iii) 2D NOESY experiments were next performed to acquire direct evidence for the double helical structure in solution. Adjacent H_f and H_g at the I-substituted phenyl rings in L,L-**IA** showed obvious NOE signals in CD₃CN, while the NOE signals between protons H_e and H_f can be attributed to the folded β-turn structure (Fig. R4). These two peaks were also observed in DMSO-*d*₆ in which L,L-**IA** exists in its monomeric form (Fig. R5), which means that the folded β-turn structure exists in DMSO-*d*₆ as well.

Significantly for L,L-**IA** in CD₃CN, distinct NOE peaks seemingly between H_e-H_g, of comparable intensity to that of the coupling of adjacent H_f-H_g, were observed (Fig. R4). This seemingly H_e-H_g coupling, however, was not observed in DMSO-*d*₆, despite the similar intramolecular β-turn structure of L,L-**IA** in DMSO-*d*₆ to that in CD₃CN (Fig. R5). This, together with the much longer *intramolecular* H_e-H_g distance of *ca.* 4.823 Å than H_f-H_g (*ca.* 2.480 Å, both distances obtained from the calculated structure of *trans*-form L,L-**IA** in CH₃CN) and the dynamic nature of the halogen-bonded double helices, leads to the conclusion that the NOE signals of the seemingly *intramolecular* H_e-H_g coupling is actually those of the *intermolecular* coupling between protons H_{e'} and H_{g'}, respectively from two adjacent *cis*-form L,L-**IA** molecules in the double helices, in which the protons H_{e'} and H_{g'} are brought by the crossed double C-I⋯S halogen bonds into close proximity of distance 2.825 Å (from the calculated structure of the oligomers of L,L-**IA** in *cis*-form in CH₃CN, Fig. R4 right).

Fig. R4 | Expanded 2D NOESY spectrum (850 MHz, 25 °C, mixing time 800 ms) of couplings between protons in phenyl rings in L,L-**IA** in CD₃CN. Calculated structures of *trans*-form L,L-**IA** and oligomers of L,L-**IA** in *cis*-form in CH₃CN, with protons and distances labeled, are shown in the right side. Protons H_e and H_g represent those in the same molecule, while H_{e'} and H_{g'} represent those from adjacent two molecules respectively. The solutions for 2D NMR measurements were sample saturated.

Fig. R5 | Expanded 2D NOESY spectrum (850 MHz, 25 °C, mixing time 800 ms) of couplings between protons in phenyl rings in *L,L*-**IA** in $\text{DMSO-}d_6$ in which *L,L*-**IA** was shown to exist in the monomer form. Molecular structure showing β -turn in *L,L*-**IA** is also presented, with protons labeled. $[\textit{L,L}\text{-IA}] = 4 \text{ mM}$.

These intermolecular $\text{H}_e\text{-H}_{g'}$ couplings were also tested for a possible single-stranded helical structure. By checking the β -turn structured *L,L*-**IA**, we proposed a right-handed single-stranded helix formed from *trans*-form *L,L*-**IA** molecules, driven by the head-to-tail $\text{C-I}\cdots\pi$ halogen bonding (Fig. R6). However, in this case the H_e and $\text{H}_{g'}$ protons respectively from the two adjacent molecules are too distant from each other by 8.687 \AA that would not lead to intermolecular NOE peaks.

Fig. R6 | Proposed right-handed single-stranded *P*-helix formed from *trans*-*L,L*-**IA** molecules as driven by the head-to-tail $\text{C-I}\cdots\pi$ halogen bonds, in which intermolecular H_e and $\text{H}_{g'}$ protons are separated by 8.687 \AA , that would not lead to intermolecular NOE couplings.

Our 2D NMR results therefore supported the supramolecular double helix structures of *cis*-form *L,L*-**IA** in CH_3CN solution.

We have added those results as Fig. 4b in the revised manuscript and Supplementary Figures 41 and 42 in the revised ESI. The relevant discussions were added and highlighted in the main text on pages 10 and 11 of the revised manuscript.

Reviewer #2 (Remarks to the Author):

I am satisfied with the authors' response concerned with my questions. Therefore, publication of this work is recommended.

Response: We thank the Reviewer for his/her recommendation.

Reviewer #3 (Remarks to the Author):

The control experiments indicate that the halide plays an important role in formation the observed supramolecular structures in solution. However, the rather long “halogen-bond” distances and the angles in the solid state makes me believe that direct spectroscopic measurements are needed to support the claims made by the authors in this manuscript.

Response: We agree with the Reviewer that direct spectroscopic measurements are needed to support the halogen-bonded double helices in solution phase. Now in the newly revised manuscript, we provide new evidence, from the similarity to the experimental CD spectrum of the calculated CD spectra of the double helical structure consisting of two and three L,L-**IA** molecules in solutions (Fig. R1), competition experiments of halogen anions at much higher concentration that suggested the contribution of the halogen bonding interactions in maintaining the double helical structures (Figs. R2 and R3), and the 2D NOESY spectra that showed NOE signals of intermolecular protons' coupling due to the double helical structures (Figs. R4-R6). These new experimental and computational results are described in detail in the response to “Reviewer 1”. The 2D NOESY spectral data provide direct spectroscopic evidence for the halogen-bonded double helical structures in the solution phase.

In addition, we also have acquired further evidence to support the halogen bonding in the solid state, using Raman and IR spectroscopy (Please refer to Figs. R8 and R9 later in this Response).

Reviewer #4 (Remarks to the Author):

Concerning the crystallography part:

most of the issues with the crystal structures analyses have been resolved. However, there are still some problems with the disordered DMF-molecules:

The authors described one of the DMF molecules with one fully occupied position (see attached figure a)). If the DMF molecules are described as a superposition of two rigid-groups with almost equal occupancies (see attached figure b)), the R1-value drops about 1% and the alerts in the checkcif-routine related to wrongly assigned H-atoms disappear.

Response: We thank the Reviewer for the excellent suggestions on the refinement of the disordered DMF-molecules. Accordingly, we have described the DMF molecules as a superposition of two rigid-groups with equal occupancies for the crystals of L,L-**IA** and D,D-**IA**, and the updated crystallographic data have been provided.

Minor issue: the table “Supplementary Table 2” contains a lot of standard uncertainties, which seem to be way too large. Please check.

Response: We have checked and revised the “Supplementary Table 2” by removing those large standard uncertainties.

General remarks:

I have some severe concerns, regarding the main claims of the manuscript:

Despite the nice illustrations (Fig.1) showing the relation to the DNA double helix, one should not forget that the similarity is of purely graphical nature and it does not reflect similarities with the bonding situation in DNA at all. In fact, in DNA the situation and therefore the energetics is completely different, which means that one cannot simply expect that both systems behave in the same manner, neither in solid state nor in solution.

Response: We agree with the Reviewer that the bonding in the here created halogen-bonded double helices is not exactly the same as that in the DNA double helix. We actually employed the structural characteristic of the DNA double helix, to extract an alternative way of building artificial double helix using a self-assembly mechanism, so to form supramolecular strands from the intermolecular non-covalent interactions, while the strands are intertwined by covalent linkages (Fig. 1a). Following this line we were able to build artificial double helices that hold some of the similarity to the DNA double helix, such as the homochiral character. This provides a new approach to artificial double helix from small molecules, which has been highlighted at the end of the “Discussion” (Page 12).

The claim of a significant contribution of halogen bonds in the crystal is not convincing in my eyes. The distances are with 3.76 Å as large as the sum of the van der Waals radii (3.78 Å) of I and S and therefore MUCH (!) larger than in “classical” systems where halogen bonds have a significant contribution (3.2-3.4 Å, see page 4 line 106-109). There are not enough details given for the DFT calculations, to judge, if even the (small) contribution of 12 KJ/mol for the halogen bonds is reliable, since there is disorder in the structure which has to be resolved and correctly be taken into account for DFT calculations.

Response: We indeed found that the I...S distances in our case are larger than in usual systems, we also noted that they are comparable to those in the classic crystals of I-containing tetrathiafulvalene derivatives or of these tetrathiafulvalene derivatives with I₃⁻ or I₂, in which the I...S distances were reported to be between 3.7 - 3.9 Å (Please refer to P. Deepa, B. Vijaya Pandiyan, P. Kolandaivel, and P. Hobza, Halogen bonds in crystal TTF derivatives: an *ab initio* quantum mechanical study. *Phys. Chem. Chem. Phys.* 2014, **16**, 2038-2047). The longer I...S distance in our case is presumably due to the crossed geometry of the two I...S contacts, in which shorter distances will lead to smaller angles, resulting in a balance between the bond lengths and angles. These descriptions are added and highlighted on page 4 of the main text of the revised manuscript.

We rechecked the original DFT calculations for the interaction energy of the C–I⋯S halogen bonding, which were based on the crystal structures without further optimization and were calculated by using the B3LYP functional. To reflect more accurately the contributions of halogen bonding, we recalculated the interaction energy by using the DFT-D3 method (wB97XD functional). Individual C–I⋯S halogen bonding energy was computed based on the dimeric structure of *cis*-L,L-**IA** abstracted from the crystal structures (as illustrated in Fig. R7), on which the positions of H-atoms were further optimized to resolve the disorder in the structure, while those of other heavy atoms were fixed without optimization to retain the C–I⋯S distance, a method that was reported in the above mentioned literature: P. Deepa, *et al.*, *Phys. Chem. Chem. Phys.* 2014, **16**, 2038-2047. The computed energy of *one* halogen bond is 23.89 kJ mol⁻¹ in the double helix containing two L,L-**IA** molecules, which is close to that calculated in the D,D-**IA** dimer (23.49 kJ mol⁻¹), supporting the reliability of the DFT calculations (Supplementary Table 3). The considerably high interaction energy can at least be partly attributed to the double crossed geometry of the halogen bonding in the double helical structure where there is considerable secondary electrostatic interaction (SEI, please refer to: W. L. Jorgensen, and J. Pranata, Importance of secondary interactions in triply hydrogen bonded complexes: guanine-cytosine vs uracil-2,6-diaminopyridine. *J. Am. Chem. Soc.* 1990, **112**, 2008-2010; W. L. Jorgensen, and D. L. Severance, Chemical chameleons: hydrogen bonding with imides and lactams in chloroform. *J. Am. Chem. Soc.* 1991, **113**, 209-216), since in the simulated dimer of unilateral analogues with only one C–I⋯S halogen bond, the interaction energy of *one* halogen bond drops dramatically to 15.04 kJ mol⁻¹ (Fig. R7).

We have added those results in the revised ESI as Supplementary Figure 3, and the relevant discussions were added and highlighted in the main text on page 4 of the revised manuscript.

Fig. R7 | Structural illustration for the DFT calculations of the bonding energies in the dimer of bilateral *cis*-L,L-**IA** with two C–I⋯S halogen bonds (a) and unilateral analogues with only one C–I⋯S halogen bond (b). The positions of H-atoms were optimized whereas those of other heavy atoms are fixed and taken from the crystal structures. Method: DFT wB97XD with the 6-31+G(d, p) basis set for C, H, O, N and S atoms, and LANL2DZ for I atom.

We also performed Raman and FTIR spectroscopic measurements to acquire additional evidence for the C–I⋯S halogen bonding in the solid state of L,L-**IA**. Compared with the synthetic starting material L-**IPhAN**₂**H**₃ having no thiourea moiety, L,L-**IA** showed a lower Raman shift at *ca.* 171 cm⁻¹ of the C–I bond (Fig. R8). This is consistent with the involvement in the halogen bonding of the I-atoms in L,L-**IA**. The stretching vibration of the C=S double bond in the thiourea moiety appeared at around 1530 cm⁻¹ (Fig. R9). We observed a red-shifted band in L,L-**IA**, compared to the control compounds L,L-**FA**, L,L-**CIA** and L,L-**BrA** that contain halogen atoms of -F, -Cl, and -Br of lower efficiency of halogen bonding, thus supporting the C–I⋯S halogen bonding in L,L-**IA** in the solid state, as that indicated in the crystal structures.

We have added these results as Supplementary Figures 6 and 7 in the revised ESI, and the relevant discussions were added and highlighted in the main text on page 6 of the revised manuscript.

Fig. R8 | Raman spectra of L-**IPhAN**₂**H**₃ and L,L-**IA** in the solid state. Inset is the chemical structure of L-**IPhAN**₂**H**₃ that has not the thiourea moiety.

Fig. R9 | Infrared spectra (KBr disk) of L,L-**FA**, L,L-**CIA**, L,L-**BrA** and L,L-**IA** in the solid state.

Especially in solution, I strongly doubt, that the weak halogen bond contribution alone is large enough to trigger the aggregation of hundreds of molecules together, forming chains of 255 nm in size. We do not have a covalently bonded backbone or the concerted formation of many hydrogen bonds leading to the formation of the double helix like we see it in DNA, since in **IA** each monomer is connected to its neighbors only by 2 weak halogen bonds. Accordingly, here no energetic gain can be obtained from the concerted formation of a special arrangement, like a double helix. This may also explain, why the authors do not see oligomers in the NMR experiments (see SI).

Response: Experiments with control molecules **HA**, **FA**, **ClA** and **BrA** indicated that the I-atoms in **IA** play a significant role in the formation of the supramolecular helical structures in solution (Figs. 3a and 3b). In the newly revised manuscript, we have now provided new evidence for the halogen bonding that maintains the double helical structures in solution phase, including calculated CD spectra of the double helical structures containing two and three **L,L-IA** molecules, that are similar to the observed CD spectrum in CH₃CN solution (Fig. R1), competition experiments with halogen anions at much higher concentration that suggest the involvement of the halogen bonding in maintaining the helical structures (Figs. R2 and R3), and more importantly, the 2D NOESY data that exhibited NOE coupling of intermolecular protons supporting the double helical structure (Figs. R4-R6). These results have been described in detail in first part of this Response to Reviewer 1.

Although the halogen bonds alone are weak, the crossed double halogen bonding dramatically stabilizes the non-covalently bonded supramolecular structures (Fig. R7). The well propagation of the helicity of the intramolecular helical β -turn fragments, in the so-formed double helical structure, would further promote the formation of highly stable supramolecular helical structures. This is supported by the observed homochiral sorting of the double helical structures in the solution phase (Fig. 5). This also explains the observation of the supramolecular helical structures in solution at low concentration of 0.08 μ M, deduced from the concentration-dependent CD spectral variations (Supplementary Figure 15 and Fig. 3d), and the high thermostability at high temperature up to 75 °C (Supplementary Figures 16-18).

I see, that the authors have carefully performed many experiments. Still, for the above reasons I cannot recommend a publication of the manuscript in the present form, because I cannot see enough evidence for the main claims, as there are:

- 1) a significant contribution of halogen bonds to the packing in the crystal and
- 2) the formation of strands composed of halogen bonded monomers in solution.

Response: We do hope that now this newly revised manuscript with new and strong evidence would be better suited for publication.

Reviewers' Comments:

Reviewer #1:

Remarks to the Author:

I am still reluctant to believe the formation of double helical conformation in CH₃CN solution based on DFT calculation and CD spectra. However, 2D NOESY data indicate the formation of different conformation from single-stranded helical structure. What was the concentration of 2D NOESY experiment? In the abstract and discussion, the authors state that "double helix forms in CH₃CN solution at an extremely low concentration (0.08 μM)", the concentration used for CD measurement. It should be restated, depending on the concentration of 2D NOESY experiment.

Reviewer #4:

Remarks to the Author:

I am truly impressed by the amount of new, obviously carefully conducted experiments and calculations the authors added to the manuscript, which seem to support a considerable contribution of the halogen bonds, responsible for the formation of a double helix of large size, both, in solid state and also in solution.

a) For the solid state, I agree based on the newly presented calculations that one might claim a significant contribution of halogen bonds to the lattice energy.

b) Still, in my eyes, neither the newly presented calculations nor the NMR, CD and Raman-spectroscopy data can confirm the presence of polymers consisting of more than two or three units in solution. Important experiments, which are common in the supramolecular chemistry community and which might give a direct proof for the existence of long polymeric chains, were not done. There are experiments as for example the determination of the binding constants from NMR titration data, ESI mass spectrometry for detection of oligomers or Scanning Tunneling Microscopy, which would be able to test the author's claims. From the new experiments presented, it seems indeed likely that in solution dimeric and maybe trimeric species exist. This scenario would fully explain the observed NMR- and CD-spectra. However, I remain very skeptical about one of the main claims, namely the formation of the polymers of larger size in solution. To the best of my knowledge, there is no other system known in literature, which is described to be capable of forming such long polymeric chains (approximately 250 nm, which corresponds to about 200 molecules) in solution, while driven only by relatively weak interactions (compared to covalent bonds) like hydrogen or halogen bonds. Even a bonding energy of 48 kJ/mol per two halogen bonds (according to the new calculations) is in my eyes simply not large enough to account for the above scenario in solution. Even more, the free energy is only a fraction of this value due to the entropy term, which is disfavors such long chains, even more so at 70 deg. C, where still chain lengths of 100 nm are claimed. Also, the DLS-experiment shows a relatively narrow size distribution at around 250 nm, which cannot be explained by a linear polymer. This points more to a cluster where a certain size is favored, due to, for example, geometrical reasons. In the case of a linear polymer, one would expect a much broader size distribution.

c) The construction of the double helix seems still very arbitrary to me. It is merely descriptive in order to establish an analogy to DNA while having no obvious energetic or chemical background. I regard the construction of the DNA-like double helix as very artificial and more confusing than illustrative. For this representation, each molecule has to be "cut" into two halves (a blue and a red one) in order to be able to draw the two strands which seem to resemble DNA. This approach is neglecting that the two halves are symmetry equivalent, related by a twofold axis. One can describe virtually any crystal structure that crystallizes in a space group with a twofold axis (or three, four and six fold) by

employing two (three, four and six) helices along this axis. In my eyes, this constructed analogy DNA-IA leads to no additional value for understanding the structure or the chemical or physical properties.

Conclusion:

As the author's claim for the existence of long polymeric chains in solution, even at high temperatures, is very spectacular and counter-intuitive, I strongly recommend to provide direct, unambiguous proof for the claims. Determination of the binding constant, ESI mass spectrometry or Scanning Tunneling Microscope investigations might be able to deliver such proof.

I want to emphasize that I consider the experimental work presented as very solid. However, in the light of the extraordinary claims one should try to be on the safe side.

Point-by-point response to the reviewers' comments

Reviewer #1 (Remarks to the Author):

I am still reluctant to believe the formation of double helical conformation in CH₃CN solution based on DFT calculation and CD spectra. However, 2D NOESY data indicate the formation of different conformation from single-stranded helical structure. What was the concentration of 2D NOESY experiment? In the abstract and discussion, the authors state that “double helix forms in CH₃CN solution at an extremely low concentration (0.08 μM)”, the concentration used for CD measurement. It should be restated, depending on the concentration of 2D NOESY experiment.

Response: Thanks for the suggestions. Indeed, concentrations of the compound taken for several experiments were not the same. We now have indicated the concentration used in each kind of experiment and accordingly revised the description in the “Abstract” on the concentration at which the double helix structures were assumed to form.

The concentration for 2D NOESY experiment was 5.6 μM, which is now indicated in the legend of Fig. 4b for the 2D NOESY spectrum. In the abstract and discussion of the revised version, we restated that “double helix forms in dilute CH₃CN solution of the micromolar concentration level, *e.g.*, 5.6 μM from 2D NOESY experiments” and “the formation of double helix at low concentration of the micromolar level, *i.e.*, 5.6 μM from 2D NOESY experiments”, respectively. These are highlighted in the main text on pages 1 and 13 of the revised manuscript.

In addition, now in this new revised manuscript, we provided further new evidence to support the polymeric chain structures in the solution phase, *i.e.*, (i) the presence of large polymeric species revealed by the ¹H NMR experiments using CH₂Cl₂ as an internal standard (Fig. R1), (ii) STM images that showed long polymeric chains of heights agreeing with those from the crystal structures (Fig. R2), and (iii) pronounced anisotropy of the hydrodynamic diameters that is indicative of the formation of long polymeric chains in solution (Fig. R3). These new experimental results have now been added in detail in the response to “Reviewer 4” and as well in the revised main text or ESI. In particular, the STM images obtained at CH₃CN-graphite interfaces showed supramolecular polymeric chain structures agreeing well with those shown in the crystal structures, adding further evidence for the formation of double helical structures in CH₃CN solution.

Reviewer #4 (Remarks to the Author):

I am truly impressed by the amount of new, obviously carefully conducted experiments and calculations the authors added to the manuscript, which seem to support a considerable contribution of the halogen bonds, responsible for the formation of a double helix of large size, both, in solid state and also in solution.

a) For the solid state, I agree based on the newly presented calculations that one might claim a significant contribution of halogen bonds to the lattice energy.

Response: We thank the Review for his/her positive impression on our new experiments and calculations. These new experiments were carried out following his/her instructive suggestions, which also strengthened our explorations of new methods for characterizing the newly formed supramolecular structures and more importantly our understanding on what was happening during the formation of those structures.

b) Still, in my eyes, neither the newly presented calculations nor the NMR, CD and Raman-spectroscopy data can confirm the presence of polymers consisting of more than two or three units in solution. Important experiments, which are common in the supramolecular chemistry community and which might give a direct proof for the existence of long polymeric chains, were not done. There are experiments as for example the determination of the binding constants from NMR titration data, ESI mass spectrometry for detection of oligomers or Scanning Tunneling Microscopy, which would be able to test the author's claims. From the new experiments presented, it seems indeed likely that in solution dimeric and maybe trimeric species exist. This scenario would fully explain the observed NMR- and CD-spectra. However, I remain very skeptical about one of the main claims, namely the formation of the polymers of larger size in solution. To the best of my knowledge, there is no other system known in literature, which is described to be capable of forming such long polymeric chains (approximately 250 nm, which corresponds to about 200 molecules) in solution, while driven only by relatively weak interactions (compared to covalent bonds) like hydrogen or halogen bonds. Even a bonding energy of 48 kJ/mol per two halogen bonds (according to the new calculations) is in my eyes simply not large enough to account for the above scenario in solution. Even more, the free energy is only a fraction of this value due to the entropy term, which is disfavors such long chains, even more so at 70 deg. C, where still chain lengths of 100 nm are claimed. Also, the DLS-experiment shows a relatively narrow size distribution at around 250 nm, which cannot be explained by a linear polymer. This points more to a cluster where a certain size is favored, due to, for example, geometrical reasons. In the case of a linear polymer, one would expect a much broader size distribution.

Response: Thanks for the comments and suggestions.

In the last version we were only able to provide the calculated CD spectra of dimeric and trimeric structures, because of the limited capacity of the computations. This could have been misleading. Actually we did note in the literature examples of long polymeric chains in solution that were driven to form by the noncovalent interactions, such as hydrogen bonding (please see Fig. 2A in the reference: Kang, J.; Miyajima, D.; Mori, T.; Inoue, Y.; Itoh, Y.; Aida, T. *Science* **2015**, *347*, 646-651). In our case, we suggested that the cooperative contributions of the intramolecularly hydrogen-bonded helical β -turn structures and the double crossed intermolecular halogen bonding are keys to support the polymeric chain structure in solution. The well propagation of the helicity of the β -turn structure corroborates the double and crossed halogen bonding in driving the chain structures. The observed self-sorting (Figure 5), among others, could be evidence to that assumption.

We appreciate very much the suggestions of the Reviewer for other methodologies to bring in new evidence for the polymeric chain structures in the solution phase. We are very much pleased that we are able to provide new

experimental data to support the chain structures in solutions, *i.e.* the ^1H NMR experiments using an internal standard, scanning tunneling microscopy (STM) images, and the pronounced anisotropy of the hydrodynamic diameters measured by DLS.

(i) The ^1H NMR signals of large polymeric species could normally be, broadened in solution NMR spectra. In our work, due to the very low achievable concentration at μM level, those broadened NMR signals are invisible as covered up by the baseline. Thus we only observed the sharp ^1H NMR signals of monomers and oligomers. To confirm the existence of large polymeric species, we introduced CH_2Cl_2 as an internal standard. According to the relative integrals of NMR signals, the percentage of monomers and oligomers are calculated to be *ca.* 12% and 5%, respectively, whereas the remaining 83% are invisible and assigned to the large polymeric species (Fig. R1).

This result supports the presence of large polymeric species in solution phase, likely not only the dimeric and trimeric species. We have added this NMR result in the revised ESI as Supplementary Figure 42 and the relevant discussions are added and highlighted in the main text on page 11 of the revised manuscript.

Fig. R1 | Partial 850 MHz ^1H NMR spectrum of $-\text{CH}_3\text{h}$ of L,L-IA and CH_2Cl_2 in CD_3CN at $25\text{ }^\circ\text{C}$. $[\text{L,L-IA}] = 5\ \mu\text{M}$, $[\text{CH}_2\text{Cl}_2] = 15\ \mu\text{M}$. CH_2Cl_2 was used as an internal standard. According to the relative integrals given in the parentheses, the percentages of monomer and oligomers are estimated to be 12% and 5%, respectively, whereas that of the remaining, with invisible signals, large polymeric chain structures is 83%.

(ii) Thanks to the Reviewer's suggestion, we explored to use STM to probe the supramolecular structures of L,L-IA and D,D-IA at the solution-graphite interface. The images show long polymeric chains (Fig. R2). The heights of the chains were determined to be 0.39 nm and 0.37 nm for L,L-IA and D,D-IA from the height profiles, agreeing well with the thickness of one *cis*-form L,L-IA and D,D-IA molecule along *b*-axis (0.35 nm), respectively deduced from the crystal structures. These results well support the 1D supramolecular polymeric chain structures in the solution phase.

Fig. R2 | STM studies. STM height images of L,L-IA (a, b, c) and D,D-IA (d) at the CH₃CN-highly oriented pyrolytic graphite (HOPG) interface, showing long polymeric chains. (e, f) The height profiles along the red dashed line in (a) and the blue dashed line in (b), respectively. (g, h) The thickness of one *cis*-L,L-IA and one *cis*-D,D-IA along *b*-axis according to the crystal structures, respectively. The CH₃CN solutions for STM experiments were diluted into 1 μM.

We have added these results in the revised ESI as Supplementary Figure 16 and in the revised manuscript as Fig. 3d. The relevant discussion were added and highlighted in the main text on page 8 of the revised manuscript.

(iii) The CH₃CN solutions of L,L-IA were prepared through sufficient ultrasonic oscillation and annealing at 75 °C to obtain thermodynamically stable polymeric species, resulting in a relatively narrow size distribution with a particle dispersion index 0.17 at the diameters around 255 nm measured from DLS experiments. We have added this explanation in the legend of the Supplementary Figure 11 of DLS experiments in the revised ESI. In particular, STM images (Fig. R2) have shown the presence of chain polymers.

Moreover, the hydrodynamic diameters of the existing species of L,L-IA in CH₃CN measured at different scattering angles varied significantly, indicating a pronounced anisotropy (Fig. R3). This is consistent with the formation of long polymeric chains of L,L-IA in the solution phase (please see the reference: Chu, B. *Laser Light Scattering: Basic Principles and Practice*, Academic Press, 1991).

We have added the results presented in Fig. R3 as Supplementary Figure 12 in the revised ESI. The relevant discussions are added and highlighted in the main text on pages 7 and 8 of the revised manuscript.

Fig. R3 | Apparent hydrodynamic diameters of L,L-**IA** in CH₃CN measured at different scattering angles θ at 25 °C. [L,L-**IA**] = 5 μ M. The significant dependence of the hydrodynamic diameters on the scattering angle suggests a pronounced anisotropy of the polymeric species. The conventional hydrodynamic diameters are measured at a fixed angle of 90°. For species of isotropic or quasi-isotropic shape, such as spherical or quasi-spherical clusters, the diameters are invariant or weakly fluctuant at different scattering angles (please see the reference: Schöpe, H. J.; Marnette, O.; van Megen, W.; Bryant, G., Preparation and characterization of particles with small differences in polydispersity. *Langmuir* **2007**, *23*, 11534-11539).

The very low concentration, *i.e.* 5 μ M, achievable in CH₃CN limited the experiments for the determination of the binding constants from NMR titrations and for identifying the oligomeric structures by ESI mass spectrometry. However, a high equilibrium constant K_e of $1.25 \times 10^7 \text{ M}^{-1}$ at 25 °C was estimated, according to the assumption that the critical aggregation concentration (*ca.* 0.08 μ M) is close to Ke^{-1} (please see the reference: Smulders, M. M. J.; Nieuwenhuizen, M. M. L.; de Greef, T. F. A.; van der Schoot, P.; Schenning, A. P. H. J.; Meijer, E. W., How to distinguish isodesmic from cooperative supramolecular polymerisation. *Chem. Eur. J.* **2010**, *16*, 362-367). The discussion is highlighted in the main text on page 8. In addition, the STM images showed long chain polymers that agree well with the supramolecular chains manifested by the crystal structures, providing evidence for the long polymeric chains in solution phase.

c) The construction of the double helix seems still very arbitrary to me. It is merely descriptive in order to establish an analogy to DNA while having no obvious energetic or chemical background. I regard the construction of the DNA-like double helix as very artificial and more confusing than illustrative. For this representation, each molecule has to be "cut" into two halves (a blue and a red one) in order to be able to draw the two strands which seem to resemble DNA. This approach is neglecting that the two halves are symmetry equivalent, related by a twofold axis. One can describe

virtually any crystal structure that crystallizes in a space group with a twofold axis (or three, four and six fold) by employing two (three, four and six) helices along this axis. In my eyes, this constructed analogy DNA-IA leads to no additional value for understanding the structure or the chemical or physical properties.

Response: In this manuscript, we actually proposed a self-assembly mechanism to artificial double helix, by forming two supramolecular strands from the intermolecular noncovalent halogen bonds, while the two strands are intertwined by covalent linkages to generate double helix. This is structurally analogous to the natural DNA double helix, while differing in that in DNA the two covalent strands are intertwined by noncovalent hydrogen bonds (Fig. 1a). An important feature we learnt from the DNA double helix was the “crossed structure”, thus we designed the compound **IAs** that allowed for double crossed halogen bonds.

Indeed “one can describe virtually any crystal structure that crystallizes in a space group with a twofold axis (or three, four and six fold) by employing two (three, four and six) helices along this axis”, these helices are however not intertwined through noncovalent or covalent interactions along the twofold axis (or three, four and six-fold axis). Our designed **L,L-IA** and **D,D-IA** molecules are of C_2 symmetry and we find a twofold axis along which two helical supramolecular strands are intertwined to generate the double helix.

Conclusion:

As the author's claim for the existence of long polymeric chains in solution, even at high temperatures, is very spectacular and counter-intuitive, I strongly recommend to provide direct, unambiguous proof for the claims. Determination of the binding constant, ESI mass spectrometry or Scanning Tunneling Microscope investigations might be able to deliver such proof.

I want to emphasize that I consider the experimental work presented as very solid. However, in the light of the extraordinary claims one should try to be on the safe side.

Response: In the revised manuscript we indicated the low concentration at which the double helix formed and the thermal stability in solution deduced from the temperature dependent CD spectra. We do hope that now this newly revised manuscript with further new evidence would be better suited for publication.

Reviewers' Comments:

Reviewer #4:

Remarks to the Author:

The authors added some more experiments, including very interesting STM images.

Some general remarks:

1) As already previously mentioned, I still consider the constructed relation to DNA as not useful in understanding the properties, structure and chemistry of this compound. In my eyes it is more misleading than helpful.

2) Concerning the formation of the claimed long polymer chains:

Even though some of the experiments might point into this direction, based on thermodynamic considerations and some unusual observations (see below), I am still not fully convinced that the compound can indeed form polymer chains of such enormous length (250 nm) in solution, even at high temperatures, and based solely on halogen bonds, as claimed.

However, I accept that it is indeed experimentally challenging to provide direct and unambiguous proof for the formation of the polymer chains in solution.

In the end, the authors are responsible for unambiguously proving their claims and discussing possible alternative scenarios, which might explain the observations, too.

So, I will not object a publication, provided that the following points are shortly addressed (either in the main text or in the SI):

1) Please introduce a short discussion about the thermodynamic situation:

- how reliable is the energy obtained from the DFT calculation? Is the value of -47 kJ/mol for ΔE realistic (compare with DFT-calculations using other functionals or give literature addressing this issue or provide model systems, where experimental values are known)?

- From the equilibrium constant given on page 8, a ΔG of -40.5 kJ/mol can be calculated. How does that value relate to similar systems or hydrogen bonded polymers or covalently bonded polymers? Actually, to my knowledge this is in the range of covalently bonded polymers. Is this value really reasonable for a purely halogen bonded system?

2) How can be explained that the length of the polymer is practically independent of the concentration?

3) How can the very narrow size distribution determined by DLS be explained, i.e. why are no shorter chains observed?

4) Is there a variation of the chain length with decreasing/increasing reaction time (ageing) in solution?

5) Why do we see only a section of the polymer chain in the STM images, and no complete chains with end points, which would allow to relate the polymer chain length to the one obtained by DLS?

6) Why is there no STM image with multiple polymer chains, as it is for example given in a previous paper of the authors (reference 22)?

7) Why are the chains so straight, compared to the bent and even loop forming DNA- and polymer-strands one can find in literature?

8) The low concentration of the compound in acetonitrile is given as a reason, why there were no mass spectrometric investigations done. This sounds very strange to me, since mass spectrometry is one of the most sensitive analytical methods one can find. Actually, if CD-spectra can be from the solution, then there should be enough material in solution to be able to get mass spectra of the compound.

Point-by-point response to the reviewers' comments

Reviewer #4 (Remarks to the Author):

The authors added some more experiments, including very interesting STM images.

Some general remarks:

1) As already previously mentioned, I still consider the constructed relation to DNA as not useful in understanding the properties, structure and chemistry of this compound. In my eyes it is more misleading than helpful.

Response: Indeed, what we really hope to show is the way we learn from the double helix structure of DNA to build an artificial double helix, which we try to demonstrate in Fig. 1a. We thus in the newly revised manuscript emphasize that the present work provides an approach towards the artificial double helices, and, also follow the suggestions of Editor, remove some of the descriptions, such as “similar to DNA” in the Abstract and in the main text.

2) Concerning the formation of the claimed long polymer chains:

Even though some of the experiments might point into this direction, based on thermodynamic considerations and some unusual observations (see below), I am still not fully convinced that the compound can indeed form polymer chains of such enormous length (250 nm) in solution, even at high temperatures, and based solely on halogen bonds, as claimed.

However, I accept that it is indeed experimentally challenging to provide direct and unambiguous proof for the formation of the polymer chains in solution.

In the end, the authors are responsible for unambiguously proving their claims and discussing possible alternative scenarios, which might explain the observations, too.

So, I will not object a publication, provided that the following points are shortly addressed (either in the main text or in the SI):

1) Please introduce a short discussion about the thermodynamic situation:

- how reliable is the energy obtained from the DFT calculation? Is the value of -47 kJ/mol for ΔE realistic (compare with DFT-calculations using other functionals or give literature addressing this issue or provide model systems, where experimental values are known)?

Response: The value of -47.78 kJ mol⁻¹ for ΔE is ascribed to two crossed C-I...S halogen bonds (Supplementary Figure 3), thus on average -23.89 kJ mol⁻¹ for one C-I...S halogen bond, which is comparable to the interaction energy calculated for I...S interaction in the classic crystal of 1,3-dithiole-2-thione-4-carboxylic acid with I₂ (*i.e.* -25.02 kJ mol⁻¹. please see Table 3 in reference: Deepa, P.; Sedlak, R.; Hobza, P., On the origin of the substantial stabilisation of the electron-donor 1,3-dithiole-2-thione-4-carboxylic acid...I₂ and DABCO...I₂ complexes. *Phys. Chem. Chem. Phys.* **2014**, *16*, 6679-6686).

We have added and highlighted the discussion in the main text on page 4 of the revised manuscript.

- From the equilibrium constant given on page 8, a ΔG of -40.5 kJ/mol can be calculated. How does that value relate to similar systems or hydrogen bonded polymers or covalently bonded polymers? Actually, to my knowledge this is in the range of covalently bonded polymers. Is this value really reasonable for a purely halogen bonded system?

Response: The value of -40.5 kJ mol⁻¹ for ΔG is comparable to that of the self-assembly of oligo(*p*-phenylenevinylene) derivatives (-39 kJ mol⁻¹ at 293 K, please see Table 2 in the reference: Smulders, M. M. J.; Nieuwenhuizen, M. M. L.; de Greef, T. F. A.; van der Schoot, P.; Schenning, A. P. H. J.; Meijer, E. W., How to distinguish isodesmic from cooperative supramolecular polymerisation. *Chem. Eur. J.* **2010**, *16*, 362-367).

The equilibrium constant of 1.25×10^7 M⁻¹ at 25 °C estimated from the concentration-dependent CD spectra is higher than those of the helical columnar stacks of classical C₃-symmetrical tricarboxamides (at 10⁵ M⁻¹ level, please see Table 1 in the reference: García, F.; Korevaar, P. A.; Verlee, A.; Meijer, E. W.; Palmans, A. R. A.; Sánchez, L., The influence of π -conjugated moieties on the thermodynamics of cooperatively self-assembling tricarboxamides. *Chem. Commun.* **2013**, *49*, 8674-8676). This is ascribed to the high cooperativity of the crossed double halogen bonds that support the double helix.

In addition, the high equilibrium constant is in agreement with the high thermal stability we observed with the supramolecular structures of L,L-**IA** in solution (Supplementary Figs. 18-20).

We have added and highlighted the discussion in the main text on page 9 of the revised manuscript.

2) How can be explained that the length of the polymer is practically independent of the concentration?

Response: A low critical aggregation concentration, 0.08 μ M, was estimated from the concentration-dependent CD spectral evolution. Therefore at the “high” concentration ranging from 0.5 to 5 μ M, stable polymeric species have already been formed, thus an increase in the solution concentration in that concentration region would mainly result in an increase in the number whereas not the length of the polymeric species.

At lower concentration, for example between 0.05 and 0.5 μ M, there might be concentration-dependent polymeric chain length. However, it turns not possible to carry out DLS measurements at those extremely low concentrations because of the sensitivity limit of DLS.

3) How can the very narrow size distribution determined by DLS be explained, i.e. why are no shorter chains observed?

Response: The CH₃CN solutions of L,L-**IA** were prepared via sufficient ultrasonic oscillation and annealing at 75 °C to obtain thermodynamically stable polymeric species, resulting in a relatively narrow size distribution. This explanation is shown in the legend of the Supplementary Figure 11 of DLS experiments and also in the Supplementary Methods (Preparation of experimental samples).

Another possible reason for this observation could be that at “high” total concentration, *i.e.* 5 μ M, most of the polymeric species are longer chains, that the shorter chains are too little to be measured at the limit of the DLS sensitivity.

4) Is there a variation of the chain length with decreasing/increasing reaction time (ageing) in solution?

Response: There is no obvious variation of the chain length with increasing ageing, as manifested by the little change in the size of the polymeric species after standing at room temperature for 2 and 7 days (Fig. R1). This could serve as a support for the stability of the halogen-bonded double helices of L,L-**IA** in CH₃CN solution.

We have added this result in the revised ESI as Supplementary Figure 23 and the relevant discussion is added and highlighted in the main text on pages 9 and 10 of the revised manuscript.

Fig. R1 | Time-dependent hydrodynamic diameters of L,L-**IA** in CH₃CN measured by dynamic light scattering at 25 °C. [L,L-**IA**] = 5 μM.

5) Why do we see only a section of the polymer chain in the STM images, and no complete chains with end points, which would allow to relate the polymer chain length to the one obtained by DLS?

Response: In order to obtain high quality STM images, the maximum scales during the STM experiments were set at 200 × 200 nm². Thus no complete chains with end points were observed.

6) Why is there no STM image with multiple polymer chains, as it is for example given in a previous paper of the authors (reference 22)?

Response: To avoid the further stacking of the polymer chains during the solvent evaporation before STM imaging, the solutions for STM experiments were diluted into 1 μM. Under this low concentration, we observe single polymer chains that are comparable to those seen in the crystal structures.

7) Why are the chains so straight, compared to the bent and even loop forming DNA- and polymer- strands one can find in literature?

Response: The straight geometry of the chains can be ascribed to the high directionality of the crossed double C–I···S halogen bonds. This is also in agreement with that shown in the crystal structures (Fig. 2c).

8) The low concentration of the compound in acetonitrile is given as a reason, why there were no mass spectrometric investigations done. This sounds very strange to me, since mass spectrometry is one of the most sensitive analytical methods one can find. Actually, if CD-spectra can be from the solution, then there should be enough material in solution to be able to get mass spectra of the compound.

Response: We indeed carried out mass spectrometric experiments and we did it again before we wrote this response. We observed MS signals of monomer and oligomers ranging from dimer to nonamer (Fig. R2), with exponentially decreased relative abundance (Fig. R3), whereas larger polymeric species were not detected. The exponential decrease in the relative abundance with the number of **IA** molecules in the oligomer, is very different from the bell-shaped molecular weight distribution of dynamic supramolecular polymers, either under an isodesmic or a cooperative self-assembly mechanism (please see Figure 1D in the reference: Smulders, M. M. J.; Nieuwenhuizen, M. M. L.; de Greef, T. F. A.; van der Schoot, P.; Schenning, A. P. H. J.; Meijer, E. W., How to distinguish isodesmic from cooperative supramolecular polymerisation. *Chem. Eur. J.* **2010**, *16*, 362-367). Despite the dependence of the relative abundance on the concentration of species is not the same from one species to the other, the observed distribution shown by our MS data (Fig. R3) does not represent the real distribution of the supramolecular polymers in the solution phase, yet likely due to the depolymerization of the polymeric species during electrospray ionization. For example, Palmans, Meijer and coworkers have employed hydrogen/deuterium exchange MS to study the dynamics of the supramolecular polymers based on the BTA motif and only the MS signal of monomer, but not those of oligomers and large polymers, was observed (“We never observed masses corresponding to dimers or lower oligomers in the mass spectra, indicating that full depolymerization of the supramolecular polymers occurred as a consequence of the ionization settings we employed”, please see page 3 of the reference: Lou, X.; Lafleur, R. P. M.; Leenders, C. M. A.; Schoenmakers, S. M. C.; Matsumoto, N. M.; Baker, M. B.; van Dongen, J. L. J.; Palmans, A. R. A.; Meijer, E. W., Dynamic diversity of synthetic supramolecular polymers in water as revealed by hydrogen/deuterium exchange. *Nat. Commun.* **2017**, *8*, 15420). In our MS experiments, we observed a small amount of oligomers, may be ascribed to the relatively higher stability of the halogen-bonded double helices, as that shown by the temperature-dependent CD spectra (Supplementary Figure 18).

Fig. R2 | ESI-HRMS spectrum of *L,L*-**IA** in CH₃CN. Inset is the experimental and simulated isotopic distribution patterns of peaks of nonamer. [*L,L*-**IA**] = 5.6 μM.

Fig. R3 | Exponential decay of the relative abundance with the number (N) of **IA** monomer in the oligomers.